



# Contrasting the Co-variability of Daytime Cloud and Precipitation over Tropical Land and Ocean

Daeho Jin[1,2], Lazaros Oreopoulos[2], Dongmin Lee[3,2], Nayeong Cho[1,2], and Jackson Tan[1,2]

[1]University Space Research Association, Columbia, MD, USA
[2]NASA Goddard Space Flight Center, Greenbelt, MD, USA
[3]Morgan State University, Baltimore MD, USA

*Correspondence to:* Daeho Jin (Daeho.Jin@nasa.gov)

**Abstract.** The co-variability of cloud and precipitation in the extended tropics (35°N−35°S) is investigated using contemporaneous datasets for a 13-year period. The goal is to quantify the relationship between cloud types and precipitation events of particular strength. Particular attention is paid to whether the relationship exhibits different characteristics over tropical land and ocean. A major analysis metric is correlation coefficients between fractions of individual cloud types and frequencies within precipitation histogram bins that have been matched in time and space. The cloud type fractions are derived from Moderate Resolution Imaging Spectroradiometer (MODIS) joint histograms of cloud top pressure and cloud optical thickness in one-degree grid cells, and the precipitation frequencies come from the Tropical Rainfall Measuring Mission (TRMM) Multi-satellite Precipitation Analysis (TMPA) dataset aggregated to the same grid.

It is found that the strongest coupling (positive correlation) between clouds and precipitation occurs for cumulonimbus clouds and heaviest rainfall over ocean. While the same cloud type and rainfall bin are also best correlated over land compared to other combinations, the correlation magnitude over land is weaker than over ocean. The difference is attributed to the greater size of convective systems over ocean. It is also found both over ocean and land that the anti-correlation of strong precipitation with "weak" (i.e., thin and/or low) cloud types is of greater absolute strength than positive correlations between weak cloud types and weak precipitation. Cloud type co-occurrence relationships explain some of the cloud-precipitation anti-correlations. Couplings between weaker rainfall and clouds are also distinct in ocean vs. land, with precipitation predictability when cloud type is known being quite poor in general, particularly over land.

## 1 Introduction

Attempts to estimate precipitation from cloud observations have a long history dating back to the era of first passive thermal infrared observations of clouds (e.g., Richards and Arkin 1981). Enlisting numerical models to help with the interpretation of observations has not been as helpful as hoped since these models generally do not produce coherent relationships between clouds and precipitation (e.g., Stephens et al. 2010; Gianotti et al. 2012; Jiang et al. 2015), with even cloud-resolving models explicitly representing precipitation processes facing challenges in that respect (e.g., Kooperman et al. 2016; Matsui et al. 2016). In the case of atmospheric global circulation model (AGCM), it is nearly impossible to resolve individual



precipitating processes due to the sub-grid nature of the problem and excessive computational burden. Hence, for AGCM evaluation, and also for observation-based water budget studies, a synoptic approach for identifying the relationships between cloud and precipitation has been deemed an inevitable compromise.

One example of employing a synoptic approach is the use of the concept of a "cloud regime" (CR) also known as "weather state" (WS; Jakob and Tselioudis 2003; Rossow et al. 2005; Oreopoulos and Rossow 2011; Tselioudis et al. 2013; Oreopoulos et al. 2014, 2016) to study precipitation characteristics. CRs represent the dominant mixtures of cloud types, and can be used as a framework to categorize cloud data in a grid (e.g., Level-3 satellite products). Using the International Satellite Cloud Climatology Project (ISCCP) WSs defined in the extended tropics (35°S−35°N), Lee et al. (2013) provided a comprehensive picture of precipitation characteristics for each WS, with an additional focus on the relationship between the most convective regime (WS1) and precipitation. Rossow et al. (2013) also conducted similar analysis but for precipitation extremes using ISCCP WSs for the deep tropical zone of 15°S−15°N. While such CR-based approaches provide valuable information about the cloud-precipitation relationship at large scales, the precipitation composites by CR encompass large spreads which obscure details of the relationship. Since CRs contain mixtures of clouds types by design, and therefore contain considerable cloud variability, ambiguities in the cloud-precipitation relationships are hard to resolve.

Cloud-precipitation relationships can, however, be examined at a more detailed level with coincident precipitation profile and cloud measurements. An example of this is the "cloud and precipitation feature database" of Liu et al. (2008) built from observations by the precipitation radar (PR), TRMM Microwave Imager (TMI), Visible and Infrared Scanner (VIRS), and Lightning Imaging System (LIS) aboard the Tropical Rainfall Measuring Mission (TRMM), and used to perform several case studies that contrasted land and ocean precipitating cloud systems. The authors showed that oceanic storms generally have larger extent at 2 km altitudes, but continental storms tend to be vertically more coherent, with a higher top and more severe rainfall. Houze et al. (2015) also reported similar results using solely vertical rainfall profiles from the TRMM PR. While these studies provide a more detailed look at the cloud-precipitation relationship thanks to the high resolution of the TRMM PR (4-5km footprint at nadir), the penalty is narrow horizontal coverage (swath widths of 215 km before orbit boost and 247 km after orbit boost).

Our paper revisits and explores once again the potential of the synoptic approach. We employ Moderate Resolution Imaging Spectroradiometer (MODIS) gridded cloud dataset and the TRMM Multi-satellite Precipitation Analysis (TMPA) dataset to explore the mesoscale cloud-precipitation relationship. While the MODIS Level-3 data are provided at 1°×1° resolution, the 2D joint histogram of cloud optical thickness ($\tau$) and cloud top pressure ($p_c$) contains pixel-level cloud information which can be combined with the sub-grid variability of precipitation at the 1° scale, available by virtue of the finer 0.25° spatial resolution of TMPA. While still coarser than TRMM PR dataset, the combined MODIS and TMPA dataset covers the entire tropics every single day, allowing better generalization of the daytime relationship between clouds and precipitation.

Our study aims to go beyond widely known cloud-precipitation associations (such as geometrically deep and optically thick clouds producing stronger rainfall), and examine instead more carefully the details of the connections between clouds and precipitation for situations that also include non-heavy precipitation. We also strive for generality of results by covering the



entire tropics and for overcoming the ambiguity of CR-based studies by taking advantage of the ability to break down the 1° cloud fraction by cloud type. We seek to answer questions such as: What are the general expectations and limitations in predicting precipitation given a cloud type in the extended tropics? Is there more close relationship between certain precipitation rates and cloud types? Do answers to the above questions differ substantially between oceans and continents?

The next section introduces the concept of "precipitation histogram" and how it can be matched and correlated to subgrid cloud type fractions at the grid level. A comprehensive examination and interpretation of cloud and precipitation co-variability over tropical land and ocean follows in Section 3. In addition to summarizing the results, the concluding Section 4 calls attention to the new insights about the nature of cloud-precipitation links that emerge from this study.

## 2 Data and Methodology

### 2.1 Cloud and Precipitation Data

Our passive cloud retrievals come from the MODIS instrument aboard the Terra and Aqua satellites. The MODIS cloud dataset (King et al., 2003) provides Level-3 cloud products at daily time scales with 1°×1° horizontal resolution. Among various cloud products, we focus on the 2D joint histogram of cloud optical thickness ($\tau$) and cloud top pressure ($p_c$). The histogram is composed of cloud fraction (CF) values along 7 classes of $p_c$ and 6 classes of $\tau$ (for a total 42 histogram bins),

and contains pixel-level cloud variability information at the 1° scale. The most recent version of the MODIS atmospheric datasets, known as "Collection 6" (Platnick et al., 2017), provides a separate histogram for "partially cloudy" (PCL) pixels, flagged as such by the so-called "clear-sky restoral" algorithm (Pincus et al., 2012; Zhang and Platnick, 2011). The PCL pixels represent usually cloud edge pixels for which the cloud property retrievals are deemed more uncertain (Cho et al., 2015). We opted to include PCL pixels in our analysis by adding the PCL histogram to the nominal histogram because, by

doing so, the MODIS cloud climatology becomes more consistent (see Oreopoulos et al. 2014) to that by ISCCP (Rossow and Schiffer, 1991, 1999), which has a long track record in cloud research and can potentially be used in a study similar to this one. As we will see later, practical considerations, and our desire to draw an analogy with the ISCCP cloud types (Chen et al., 2000; Rossow and Schiffer, 1999), led us to coarsen the histogram from 42 to 9 bins.

The precipitation dataset used in our study is the 3B42 research product (version 7) of Tropical Rainfall Measuring Mission

(TRMM) Multi-satellite Precipitation Analysis (TMPA) (Huffman et al., 2007, 2010; Huffman and Bolvin, 2015). The TMPA pursues the "best" satellite precipitation estimates using TRMM Microwave Imager (TMI) and Precipitation Radar (PR) data as calibrators in merging measurements from several microwave and infrared sensors, and monthly gauge data (over land) from the Global Precipitation Climatology Centre (GPCC; Huffman et al. 2007). The horizontal resolution of TMPA is 0.25°×0.25° covering 50°S to 50°N. TMPA is available from January 1998 with 3-hourly resolution, but we use

the period from December 2002 to November 2015 which overlaps temporally with Aqua and Terra MODIS data. Since we pursue the co-variability of cloud and precipitation, and one of the essential pieces of cloud information is the optical thickness which is only available during daytime, our study relies on measurements only around the Terra and Aqua



overpasses of 10:30 am and 1:30 pm local solar time (LST), respectively. We restrict our study to the extended tropical region (35°N − 35°S) to avoid ambiguities in the interpretation of the MODIS joint histograms which include progressively more temporal variability towards higher latitudes as data from successive spatially overlapping orbits fall within the same 1° grid cell. Still, we should note that when various aspects of the analysis were tested on the full TMPA spatial coverage

(50°N − 50°S), the results were not substantially different. Lastly, since it is well-established that precipitation properties over land and ocean are quite different (e.g., Williams and Stanfill 2002; Zipser et al. 2006; Matsui et al. 2016), we maintain via the MODIS land-water mask (Carroll et al., 2009) distinct land and ocean results throughout our analysis. At the 1° resolution, a grid cell is marked as ocean when the water mask area is greater than 90%, while it is marked as land when the water mask area is smaller than 10%. For our extended tropics domain this definition assigns 71.1% of the grid cells to the

ocean and 24.1% to the land category.

The quality of the TMPA product differs between land and ocean, mainly due to two factors: (1) Gauge adjustment which reduces systematic biases in land precipitation, and (2) Satellite retrieval algorithm differences which result in lower random errors over ocean (Liu, 2016; Sapiano and Arkin, 2009; Tian and Peters-Lidard, 2010). We assert that our findings about ocean-land differences are not much affected by these algorithmic variations because, first, random errors should be

suppressed due to large sample size, and second, our analysis is largely based on deviations from the mean state. Nevertheless, it is conceivable that TMPA overall has less reliable data in certain situations such as continental warm rains (Kidder and Vonder Haar, 1995; Kummerow et al., 2015).

### 2.2 Matching Precipitation Data to Cloud Grid

The 3B42 dataset is of higher resolution than that of MODIS Level-3 cloud data, and we therefore resample the 3B42 data to

the 1°×1° resolution of the MODIS dataset. Previous studies averaged precipitation rates to a single value representing grid mean (e.g., Lee et al. 2013; Rossow et al. 2013). In this study, a marginal histogram of 3B42 0.25° grid precipitation rates is created for each 1°×1° grid cell. The idea of such 1° precipitation histograms was drawn from our other main data set, the MODIS joint 2D histogram of $p_c$−$\tau$, which preserves a certain degree of sub-grid cloud information (although not of the spatial distribution of the sub-grid variability). So, in a sense, sub-grid information about precipitation rate can also be

preserved in the form of histogram by assigning the 16 (when no missing values exist) values of precipitation rate at 0.25°× 0.25° resolution to pre-defined bins to create a marginal histogram at 1° grid cell. The histogram is normalized by dividing each bin count by the total count in the histogram bins, i.e. 16 if no missing. If the number of bins in the histogram is chosen to be also 16, each bin value falls between 0 and 1 in multiples of 1/16, the sum of all histogram bins at 1° grid cell is equal to 1, and sub-grid precipitation rates are thus converted to *areal fractions* of specific ranges of precipitation rates. One of the

16 precipitation histogram bins corresponds to "no-rain" and the remaining 15 bins to rain rates greater than zero. Histogram bin boundaries are selected with fifteen logarithmically-spaced intervals to ensure a more even distribution of counts (see Fig. 1). Figure 1 shows distribution of precipitation rate of the original TMPA data in our domain of the extended tropics according to this histogram binning approach. We see that the amount of missing data is negligible, and that the "no-rain"



bin has an 89.5% share of all data points. The rain rate around 1 mm/hr has a maximum share near 1.1%, and extreme values are below 0.4% at both low and high rain rates.

In addition to the trivial matching of grid cells, the time of the TMPA and MODIS observations also needs to be matched. Since MODIS Level-3 cloud data come from the aggregation of retrieved satellite observation along the Terra or Aqua paths,

and since these satellites are in a sun-synchronous orbit, each grid cell of a daily MODIS map has a limited range of nominal LST, but has a varying Coordinated Universal Time (UTC), the time keeping system of TMPA. The UTC of each grid cell can be estimated from the mean solar zenith angle (SZA) available as a MODIS Level-3 variable. Because of minimal overlap of satellite orbits in the tropics, the mean SZA value is a result of mostly (small) spatial variations within the 1° grid cell. After identifying the UTC corresponding to the grid cell of cloud data, the proper TMPA data can be extracted. Since

the 3B42 data is available at 3 hour-intervals, the maximum time difference between cloud and precipitation observations is 1.5 hours in this study.

The histograms of TMPA tropical rainfall rate that matches Terra and Aqua paths spatially and temporally are also shown in Fig. 1. One notable change from the original TMPA data to Terra- or Aqua-matched data is that the portion of missing data now surges to over 5% of total data points. Most of these missing data are traced back to unavailable Level-3 MODIS data,

for reasons such as absence of clouds or gaps between consecutive Terra-Aqua orbits at low latitudes. In explaining other differences in frequencies between original and matched data, the effect of the diurnal cycle should be considered. For example, at the Terra overpass time of around 10:30 am (LST), precipitation is relatively suppressed over both land and ocean (e.g., Yang and Smith 2006; Kikuchi and Wang 2008). This appears in Fig. 1 as Terra-matched precipitation having smaller frequencies than the original and the Aqua-matched precipitation. It is also notable that, for weak-to-moderate

precipitation rate (less than 1mm/hr), even Aqua-matched precipitation is (slightly) lower percentage-wise than fully-sampled TMPA precipitation.

**2.3 Analysis Method and Simplification of Cloud and Precipitation Histograms**

The simplest and most straightforward method to measure the co-variability of two variables is to calculate their cross-correlation coefficients, namely Pearson's $r$. In this study, the values in each bin of the $p_c$-$\tau$ joint histogram and of the

precipitation histogram form large arrays (O(1,000,000)) in the spatio-temporal domain, from which we can calculate the correlation coefficient between cloud and precipitation bin values as time and location varies. The original resolution of the $p_c$-$\tau$ and precipitation histograms yields 672 (= 42 CF bins × 16 precipitation bins) correlation coefficients. This is simply too large a number of coefficients to analyze, visualize, and make sense of. Hence, a coarsening of both histograms before correlations are calculated is in order per the procedure described below.

First, the cloud histogram bins are reduced from 42 bins to the 9 ISCCP cloud types defined in Rossow and Schiffer (1999). Figure 2 shows the $p_c$ and $\tau$ range for each cloud type. Low and mid-level cloud types are composed of 4 CF bins (= 2 $p_c$ classes × 2 $\tau$ classes) while high cloud types are composed of 6 CF bins (= 3 $p_c$ classes × 2 $\tau$ classes). Hence, the CF value of each cloud type comes from the summation of either 4 or 6 CF bin values of the original 2D joint histogram. Similarly, the



16 histogram bins of precipitation are reduced to 6 groups. The "no-rain" bin is unchanged, and the other 15 bins of measurable rainfall are resampled to 5 precipitation groups (each called as a "P-group" hereafter) by summing three consecutive precipitation bins, as shown at the bottom of Fig. 1. Each P-group is labelled from P1 to P5, with P1 representing the lightest precipitation, and P5 representing the heaviest precipitation. Henceforth, for simplicity, the same

symbols are also used for the frequency of occurrence within these groups, with no confusion resulting, as the context is invariably clear. As a result, the number of correlation coefficients decreases to 54 (= 9 cloud type CF values × 6 P-group frequencies). We also note that the Terra and Aqua data (and matched precipitation data) are considered as a single ensemble, so our results represent the local cloud-precipitation co-variability for the 6-hour daytime period spanning 1.5 hour before the Terra overpass and 1.5 hour after the Aqua overpass.

**3 Land-Ocean Difference of cloud-precipitation relationship**

**3.1 Basic Statistics and Composite Means of Cloud and Precipitation Data**

Before examining correlations between cloud and precipitation data, it is illuminating to examine the basic statistical information and mean states of both histograms from which co-variability of anomalies is extracted. First, we examine the P-groups that co-exist with certain cloud fractions at the grid-level. Figures 3 and 4 show the conditional probability of P-group

occurrence under the condition that a particular cloud type exists over ocean (Fig. 3) and land (Fig. 4). For example, for all oceanic 1° grid cells with Cumulonimbus (Cb) clouds occurring, about 52% of the grid cells report P5 precipitation at one or more 0.25° sub-cell(s) (Fig. 3a, upper-right bin). The threshold CF that determines cloud occurrence is set to 6.25%, i.e. the same threshold fraction (1/16) that defines precipitation occurrence. We note that P-groups are not mutually exclusive because several P-groups can occur simultaneously in a 1° grid cell.

Over ocean, the cloud type co-occurring the most with precipitation rates of medium to heavy intensity is Cb. The P-group most likely to occur alongside Cb clouds is P4 with a probability of 0.77 (Fig. 3b). The probability of P5 group occurrence is lower at 0.52, but also comes with an overall P5 population smaller than that of P4 (Fig. 1 and Table 1). When precipitation of any intensity is considered (Fig. 3f), besides Cb having the highest probability of precipitation, 0.90, oceanic Nimbostratus (Ns) also emerges with a high probability of 0.75. The no-rain occurrences are, not surprisingly, better

associated with thin and/or low clouds (so-called "weak" clouds), topped by the 0.82 probability for Cumulus (Cu) clouds. It is notable that no-rain probabilities are clearly distinguishable from those of the weak P1 or P2 rain groups not only by the probability of these P-groups occurring (we note that the population of no-rain case is much larger), but also by how the probability varies with cloud type within the precipitation group (e.g., compare Cu and Ns as an extreme contrast). Comparing Figs. 3 and 4, we see that land clouds generally have a smaller chance of precipitation co-existing with clouds at

the 1° scale. Even the P4 precipitation probability of Cb clouds is only 0.54 (Fig. 4b), far lower than its oceanic counterpart of 0.77. For the case of rainfall with any intensity (Fig. 4f), the precipitation probability of Ns is only 0.35 compared to 0.75 over ocean. The precipitation probability of mid-level Altostratus (As) also decreases from 0.53 to 0.31, so mid-level clouds



seem particularly less active precipitation producers over land. In addition, the lightest rain group P1 over land is not associated with any particular cloud type (Fig. 4e vs. Fig. 3e) while the no-rain case exhibits strong probability dependence on cloud type. The issue of less rain over land will be revisited later.

Figures 5 (ocean) and 6 (land) show composite mean cloud and precipitation histograms, for occurrences of the strongest precipitation groups P5 and P4 (i.e., at least one of the subgrids within the 1° grid cell has a precipitation rate of P5 or P4 class, respectively). When P5 occurs over ocean (Fig. 5), both cloud and rainy fractions are larger compared to the P4 cases. On the cloud side, Cb exhibits the largest increases in CF when moving from the P4 to the P5 composite. For the P5 composite, the largest CFs (red color) are located in the bins with $p_c$ below 310 hPa and the $\tau$ bins extending from 9.4 to 60, while in the P4 composite, CF peaks in the bin bounded by 310 and 180 hPa, and with $\tau$ between 3.6 to 23. Conversely, thin cloud CFs as well as stratocumulus (Sc) CF are smaller in the P5 composite than the P4 composite. However, it cannot be determined from this analysis alone whether the increased amounts of thin and Sc clouds in the P4 composite are directly linked with the occurrence of P4 precipitation, or are a consequence of increased chance of co-existence with other clouds producing P4 precipitation. The CFs of mid-level clouds increase only slightly from P5 to P4 composites in terms of absolute values, but these increases are quite large in a relative sense because absolute CF values for these clouds are very small in the MODIS climatology.

Consistent with the CF changes, the total rainy fraction, which is defined as the fraction sum of the 15 precipitation histogram bins excluding the "No-rain" bin in 1° grid cells, also increases in the P5 composite (0.794 vs. 0.627). The mean precipitation histogram in the P5 composite (Fig. 5 top right) shows that the peak occurs within the P5 group, but the fraction of total precipitation due to the P4 group is larger. This does not come as a surprise because, first, the absolute population of P4 is higher than that of P5 (Fig. 1) and second, most P5 precipitation events co-occur with P4 precipitation events at 1° resolution (Table 1). The P4 fractional contribution in the P5 composite is also larger than the P4 contribution in the P4 composite (Fig. 5 bottom right), while the light to moderate P-group (P1-P3) fractions are slightly larger in the P4 composite compared to the P5 composite. This indicates that stronger precipitation events also have a weaker tail towards lower rainfall rates. In terms of total rainy fraction, considering that approximately 38% of the P4 composite population overlaps with the P5 composite (Table 1), we see that the horizontal sizes of oceanic rain systems producing P4 but not P5 are often much smaller than systems producing P5.

We also examined the geographical distributions of P4 and P5 occurrence frequency (supplementary Fig. 1), and found that the distribution maps look very similar, with the regions significantly skewed towards one of P4 or P5 being very few. This result suggests that the P4 and P5 composites in Fig. 5 are related and likely capture the developing and mature stage of mesoscale convective systems (MCSs), respectively. The review of Houze (2004) and Chapter 9 of Houze (2014) describe MCS as the combined system of a large region of stratiform precipitation paired with individual or clustered Cb clouds, yielding thus a variety of cloud and precipitation structures (Houze et al., 1990). The P5 composite patterns of cloud and precipitation shown in Fig. 5 are in accordance with such MCS characteristics.



Figure 6 shows the same P4 and P5 composite means as Fig. 5, but over land. Comparing the top and bottom panels of Fig. 6, we see that the general characteristics of the differences between P4 and P5 land composites are similar to their oceanic counterparts. For example, the total CF and rainy fraction increase from the P4 to the P5 composites, accompanied by larger CFs of Cb clouds, and P4 group fractional contribution in the P5 composite. However, there are also notable differences, such as total CF difference between P4 and P5 composites being smaller over land than over ocean. Over land, smaller CFs can produce P5-magnitude precipitation while larger CFs are needed for P4-magnitude precipitation compared to ocean. The total rainy fractions of P4 and P5 composites are also smaller over land. For example, when P5 occurs, 79% of oceanic sub-cells at 1° scales are precipitating, while the same is true for only 59% of continental sub-cells. For P4 composites, the values are 63% over ocean vs. 47% over land. These are strong indications that continental systems producing heavy rainfall are in general smaller in size than their oceanic brethren (Liu et al. 2008; Houze 2014, Chapter 9; Houze et al. 2015).

The distributions of total rainy fraction as well as grid mean cloud properties by P-group are further examined in Fig. 7, which shows boxplots of total rainy fraction, CF, $\log10(\tau)$, and $p_c$ distributions. Over ocean, both total rainy fraction and CF generally increase linearly with precipitation. However, the picture is somewhat different for land. From P2 to P5, both the median and mean values of land CF are quite similar (Fig. 7b). As a result, in the P2 case, the land CF median is nearly 10% greater than the ocean CF median, while it is 5% smaller than its ocean CF counterpart in the case of P5. At the same time, the total rainy fraction over land appears to be linearly increasing in the same way as over ocean, albeit with a notably smaller absolute value and slope of growth. Hence, it appears that over land, similar amounts of CF (e.g., 70% to 80%) in a 1° grid cell are involved in a broad range of precipitation rates, while the fraction of raining clouds in the grid cell is much smaller compared to ocean. Collectively, these results indicate reduced predictability of precipitation from knowledge of CF over land.

Houze (2014, Chapter 9) and Houze et al. (2015) noted that shallow and isolated clouds producing "warm rain" are mostly oceanic phenomena, while the size of MCSs is generally larger over ocean than land. These two different precipitating sources can explain the big contrast over ocean of total rainy fraction between P2 (median 35%) and P5 (median 85%); over land the difference is less than 30%. In addition, Figs. 7c and 7d show that light-to-medium precipitation groups (P1-P3) over ocean are associated with optically thinner and shallower clouds, another evidence of the prevalence of marine "warm" rain processes. Note also the larger variability (taller inter-quartile box) of oceanic $p_c$ for light precipitation compared to heavy precipitation, indicating that the former is harder to relate to particular cloud types. For continental light precipitation, associated clouds are optically thicker and of higher altitude than oceanic counterparts, but TMPA's potential weakness in identifying it, as described in section 2.1, can be a factor affecting Fig. 7 results.

In summary, the 1° spatio-temporally matched cloud and precipitation data suggest that prevailing features such as contrasting horizontal size of oceanic and land MCSs can be clearly detected by this study's methods. In the next subsection, the covariability of cloud and precipitation is examined in detail using explicit correlation analysis.



### 3.2 Correlations between Cloud and Precipitation Fractions

As stated previously, to measure the co-variability of cloud and precipitation, we calculate cross-correlation coefficients between the CFs of the 9 cloud types and the normalized frequencies (equivalent to fraction of precipitating area with our binning conventions) within the 5 P-groups. Figures 8 and 9 show correlations of cloud types for each P-group as well as all

combinations of consecutively cumulative P-group frequencies over the oceanic and land regions of our extended tropical domain. We note that, when the fraction sum of specific P-group(s) is zero, the data point is excluded from the calculation of correlations. Hence, for example, correlation coefficients with P5 over ocean (Fig. 8a) are calculated with approximately 4.5% of the total data available. Even over land, the sample size for this case (Fig. 9a) nevertheless exceeds 1 million, placing the 99% significance level to less than 0.005 in correlation coefficient absolute value. The statistical significance

level was calculated here using a bootstrapping method which randomly shuffles the array, but in a way that considers the effect of autocorrelations between neighboring grid cells (i.e., shuffling by "chunks"; Kunsch 1989; Léger et al. 1992; Liu and Singh 1992). With the significance level quoted above, all correlations in Figures 8 and 9 are statistically significant.

Looking first at oceanic cloud-precipitation coupling, Fig. 8 reveals that strong correlations, both negative and positive, occur in the panels on the left, whilst correlations weaken as one moves to the right. The leftmost column panels consist of

P-group(s) that include P5, the group of heaviest precipitation, while as one moves to the right, heavier precipitation is progressively excluded. The overall picture then is that strong correlations correspond mostly to heavy precipitation while light precipitation correlates poorly with any cloud type. The leftmost column panels of Fig. 8 indicates that positive coefficients occur for high cloud types of moderate to strong optical thickness, namely Cirrostratus (Cs; probably includes many anvils) and Cb (deep convection core), while negative values occur for low cloud types that are also optically thin. In

the 5 panels of the leftmost column, Cb clouds always have strong positive correlations with precipitation, a result that should not come as a surprise. For the correlation of Cs clouds to become positive and then increase, lighter precipitation has to be added to P5. For example, when only P5 values are used (Fig. 8a), the correlation coefficient of Cs clouds is negative (-0.16), but changes to 0.22 after P4 is added to P5 (Fig. 8b). This suggests that lighter rain in the vicinity of the heaviest rain is more closely related to Cs (or anvil) clouds. A similar trend of stronger correlations when lighter precipitation is included

also ensues for low and thin Cu clouds, although with a negative sign in this case. In Fig. 8a, a strong negative correlation is seen with (high and thin) Cirrus (Ci) clouds, and as lighter precipitation is added, the peak of negative correlations moves towards lower Cu clouds.

Notable patterns in correlation coefficients are also detected in the second left column panels which show correlations with P4 precipitation included, but not P5. Similar to the leftmost column panels, Cb, Cs, and Cu clouds show the stronger

correlations with positive or negative signs. One difference from the P5 cases is that, in Figs. 8e, 8h, and 8l, the positive correlations of Cs clouds are stronger than those of the thicker Cb clouds. The correlation coefficient values of Cs clouds in these panels are quite similar to the values for the same clouds in the leftmost column (which includes P5 precipitation). This result suggests that it is actually Cs clouds that are related the most to the variability of P4 and lighter precipitation.



Moving now to the land regions of our extended tropical domain, we use the same "correlation pyramid" to note that the relationship between high and optically thick Cb clouds and P5 heavy precipitation is of positive strength similar to that over oceans (Fig. 9). However, other details are quite different between land and ocean. First, the negative correlations of Cu clouds in the leftmost column panels are weaker. In Fig. 8, the peak negative correlation values reached -0.40 and occurred

in panels (d) and (g) which include the moderate to weak precipitation of the P3 and P2 groups. In Fig. 9 on the other hand, the peak negative value is a weaker value of -0.23 and is seen in panel (b), which represents the sum of only P4 and P5 precipitation; the negative correlations weaken as lighter precipitation is added. This result suggests better chances of Cu clouds and P5 precipitation co-existing in 1° grid cells over land compared to ocean. This observation may be related to our earlier finding drawn from Figs. 5, 6 and 7 that the size of raining systems is much smaller over land than ocean.

Secondary but still noteworthy differences between land and ocean are identified in the correlations between Cs clouds and precipitation that includes the P4 and P3 groups (second and third column panels from left). Previously in Fig. 8, the maximum correlation values in the second-from-left column were the ones correlated with Cs clouds, up to 0.36. In the third column associated with P3 precipitation, correlations with Cs clouds weaken to 0.16. In contrast, Fig. 9 shows that the strongest correlations of the second column are those for Cb clouds, not Cs. In the third column, the correlations with Cs

clouds are not weakened as much, with a 0.21 correlation value being reached in P-groups that include P3. This pattern indicates that continental high clouds are better correlated with lighter precipitation. It is also notable that correlation coefficients with Cs clouds in the first column of Fig. 9 (including P5 over land) reach just 0.25, while those in Fig. 8 (ocean) are as high as 0.39. A possible explanation of the above correlation results is the possibility that thick anvils of continental MCS (Cetrone and Houze, 2009; Yuan et al., 2011) are more frequently classified as Cb rather than Cs by the

definition of cloud types in this study.

For light precipitation the absolute values of correlation coefficients are smaller than those for heavy precipitation commonly found over land and ocean, reflecting the fact that the mechanisms and origins of light precipitation exhibit a greater variety. Nevertheless, a meaningful difference between land and ocean can be seen in Figs. 8 and 9. When comparing Figs. 8n and 8o with 9n and 9o, peak correlations around 0.1 occur for Stratus (St) over ocean, but such values remain with Cs and Cb over

land. This result suggests that over land even light precipitation is more frequently related to strong convective activity while oceanic light precipitation has a greater chance of being produced by "warm rain" mechanisms, as noted at the end of subsection 3.1.

In summary, continental Cs and Cb clouds co-exist with a broader range of precipitation, but are also more weakly correlated with them, compared to their oceanic brethren. This result is consistent with the previously noted climatological features of

grid mean cloud properties shown in Fig. 7. For example, the median $p_c$ for the P2 group over land in Fig. 7d is already below 440hPa, while for oceanic clouds the median $p_c$ reaches such values when precipitation is strong enough to belong to the P4 group. The optical thickness is also generally larger for land clouds (Fig. 7c). It is possible that TMPA is missing some "warm" rain events over land due to microwave retrieval inadequacies as stated in subsections 2.1 and 3.1. For heavy precipitation, Level-2 TRMM observations led Liu et al. (2008) to conclude that tropical land storms are more vertically



developed, i.e. optically thicker clouds with higher tops, but also horizontally smaller than oceanic storms (see also Houze et al. 2015; Matsui et al. 2016). Hence, precipitation over land is spread over a smaller area, resulting in weaker correlations at the level of one-degree areas. Differences in correlations between Figs. 8 and 9 are therefore likely good indicators of land/ocean differences in the nature of tropical storms or MCSs.

There are other intriguing aspects of cloud-precipitation co-variability in land and ocean, and these are examined more closely in the next subsection: (1) the origin of negative correlations and (2) correlation sensitivity to precipitation strength.

### 3.3 Further Investigation for Correlation Features

#### 3.3.1 Negative correlations between precipitation and thin clouds

In Figs. 8 and 9, we found that thin clouds have negative correlations with heavy precipitation. These negative correlations can be interpreted as thin clouds being rarer when heavy precipitation occurs, an interpretation that is consistent with empirical observation and expectations. However, since it is also seen that heavy precipitation is strongly related to thick and high-level clouds (e.g., Cb), the negative correlation of optically thin clouds with heavy precipitation can also be interpreted as a contemporaneous negative co-occurrence relationship between optically thin low and optically thick high clouds. Please note that for a cloud type to be always (i.e., regardless of precipitation strength) anti-correlated with precipitation, its occurrence must be anti-correlated with that of other cloud types that are positively correlated with precipitation of a certain range. In order to examine these issues, we calculate internal correlations among cloud types based on the spatiotemporal variability of their CFs. From all possible combinations, we elect to show results where one of the cloud types is either Cs or Cb when P4 or P5 precipitation group occurs. These results for both land and ocean are shown in Fig. 10. For example, Figs. 10a (ocean) and 10d (land) show correlation coefficients between the CF of Cs and the CFs of all other cloud types for grid cells reporting P4 precipitation. We note that the samples used for Fig. 10 correlations are the same samples used for cloud-precipitation correlations shown in Figs. 8 and 9, if the condition, namely P4 or P5 values greater than zero, is the same (e.g., Figs 10a, 10b, and 8c).

Figures 10a, 10b, and 10c show correlation coefficients based on oceanic Cs and Cb CFs. The Cs clouds are strongly anti-correlated with Cu and Sc clouds, while Cb clouds are furthermore also strongly anti-correlated with Ci clouds. In the cases when P5 precipitation is detected (Fig. 10c), the anti-correlation between Cb and Ci CFs becomes even stronger. Actually in this case, Cb clouds are anti-correlated with all other cloud types; i.e., when Cb CF increases at 1° grid cell, CFs of other clouds decreases, and vice versa. These cloud type correlation patterns remind us of Figs. 8a, 8b, and 8c. For example, a comparison between Figs. 10c and 8a shows that the anti-correlation ordering by strength is the same, with Ci clouds coming first, Cu second, and Sc clouds third. This result suggests that in tropical oceans P5 precipitation is mainly related to Cb clouds, and its anti-correlation with thin clouds is another expression of the anti-correlation between Cb and thin clouds. The exact origins of this anti-correlation are unknown because a passive sensor such as MODIS is greatly limited in



distinguishing between cases where the mid- and low-level clouds are absent and cases where they are obscured by high clouds.

When focus is shifted to the weaker P4 precipitation class, both Cs and Cb clouds are anti-correlated with low Cu and Sc clouds with only slightly weaker correlation for Sc than Cu (Figs. 10a and 10b). Previously however, Fig. 8c indicated that

the anti-correlation between P4-class and Sc cloud is much weaker than that between P4 and Cu cloud (-0.15 vs. -0.28). This discrepancy is also seen in all panels of Fig. 8 representing correlations with moderate to heavy precipitation classes (three columns from left), but is not seen over land (Fig. 9). This issue will be discussed further in the next subsection which deals with correlation sensitivity, but suffice it to say here that cloud-precipitation anti-correlations cannot be exclusively attributed to cloud type co-occurrence anti-correlations.

Comparing oceanic and continental correlation patterns in Fig. 10 (top row vs. bottom row), the correlation patterns are quite similar, but with weaker correlation magnitudes over land. For example, Cs clouds in Fig. 10d remain strongly anti-correlated with Cu and Sc clouds, and Cb clouds in Fig. 10f are still anti-correlated with all other cloud types. Yet, differences between ocean and land clouds also emerge. First, particularly in the presence of non-zero P4 precipitation (Figs. 10d and e), there are stronger anti-correlations between Cb or Cs clouds and mid-level clouds over land. Previously in Fig. 6,

we noted that mid-level clouds have greater CFs over land compared to ocean (even though their absolute value is much smaller than high clouds). The increased CFs of mid-level clouds over land may be related to a closer relationship with high-thick clouds, thus affecting the correlation strength.

Another difference between ocean and land is the correlation between Cb and Cs in the presence of P5-class precipitation. Comparing Figs. 10c and 10f, the notable anti-correlation value of -0.27 over ocean weakens to -0.16 over land. This result

indicates that Cs and Cb clouds are less mutually exclusive over land. Since the overcast condition (100% CF) in a 1° grid cell is more frequent over ocean (Fig. 7b), indicating that oceanic MCS can grow to sizes larger than 1°, there is a greater chance of competition between Cs and Cb over ocean to fill the grid cell.

Lastly, we return to our previous point that the anti-correlation of CFs among cloud types does not explain all features of the anti-correlation between cloud and precipitation shown in Figs. 8 and 9. For example, comparing Figs. 8a and 9a, anti-

correlation between P5 and Cu cloud weakens from -0.25 (ocean) to -0.20 (land). However, Figs. 10c and 10f indicate that the anti-correlations between Cb and Cu clouds are almost the same for ocean and land (-0.37 vs -0.36). This further supports the hypothesis that the frequencies of the P5 precipitation group and Cb CFs are more weakly coupled over land.

### 3.3.2 Correlation sensitivity to heavy precipitation

Correlations between cloud and precipitation shown previously in Figs. 8 and 9 indicated that the heaviest precipitation

group has a solid relationship (correlation or anti-correlation) with cloud types, while weaker precipitation groups do not. This fundamental finding is examined more closely with more detailed CF-precipitation correlations. Figure 11 shows correlation coefficients over both ocean and land between CF of various cloud types and frequency of cumulative precipitation within original precipitation bins, from the 7th bin onward (i.e., 0.251 mm / hr and above). Hence, at the start of



the x-axis the precipitation frequency corresponds only to the 7th bin, and as one moves along the axis precipitation frequencies for subsequent bins are progressively added until the end of the axis where the precipitation frequency represents the sum of all values from the 7th to 15th bin, namely the sum of frequencies of the P3, P4 and P5 groups. When compared to Figs. 8 or 9, Fig.11 shows essentially in more detail the evolution of correlation coefficients for the third row of the

"pyramid", i.e. correlation changes as one moves from Fig. 8f (9f) to 8e (9e) and then to 8d (9d) over ocean (over land).

Figure 11a shows the correlation change of high clouds (Ci, Cs, and Cb). Over ocean (solid line), the correlation of Cb cloud increases monotonically as heavy precipitation is added, while that of Cs cloud peaks when the 13th bin (2.51−3.98 mm/hr) is added; further additions of heavier precipitation results in correlation coefficients trending downward. Similar patterns are also seen for the land clouds in this category. However, one prominent difference between ocean and land is that the land

clouds in this group tend to be more strongly correlated with weaker precipitation. For example, continental Cb clouds correlate better than oceanic Cb to precipitation up to the 13th bin. But the correlation curve for oceanic Cb clouds exhibits a steeper slope after the 11th bin has been added, and ends up surpassing continental Cb clouds with the heaviest precipitation. In the case of Cs cloud, the continental correlation curve peaks with the addition of the 11th bin (1.0−1.58 mm/hr), while the oceanic peaks when the 13th bin is added. This result indicates that P4 precipitation over land tends to be more related with

Cb than Cs clouds, which contrasts what happens over ocean. In the case of Ci clouds, the anti-correlation is stronger at weak precipitation over land, consistent with the above argument, but the difference between land and ocean is not very pronounced given the small absolute values of coefficients compared to Cs and Cb clouds.

For the mid-height cloud group shown in Fig.11b, a notable difference between ocean and land is seen for the As and Ns clouds. Oceanic Ns clouds have broad positive correlations around the value of 0.1 for all precipitation bins. Oceanic As also

have positive correlations with moderate-to-heavy precipitation bins, although they decrease to zero as the heaviest precipitation is added. On the other hand, continental As and Ns clouds show only negative correlations with all precipitation strengths. As and Ns occurrences are smaller over ocean than over land ([3.8%, 1.7%] ocean vs. [5.3%, 2.5%] land in Figs. 5 and 6, P4>0), but shallower convection over ocean seems sufficiently strong to produce moderate-to-heavy precipitation from As and Ns clouds.

In the case of the low cloud group shown in Fig.11c, first, the thickest St cloud's correlation evolution pattern looks similar to that of As cloud above, although the presence of St cloud over ocean is even smaller than As (St CF=1.3% vs. As CF=3.8% when P4>0 in Fig. 5). The correlation pyramid of Fig. 8 has shown that the positive correlation of St cloud is stronger when it is related to weak precipitation classes (P1 or P2) which are not included here (see Supplementary Fig. 2). Secondly, also notable is the contrasting correlation evolutions of oceanic Cu and Sc clouds, previously mentioned to have

different magnitudes of anti-correlation. Oceanic Sc clouds have slightly positive correlations with the 7th and 7th-to-8th precipitation bins which then become negative as heavier precipitation is added. Similar to the St cloud, the positive correlation of Sc cloud is expected to strengthen with even lighter precipitation (Fig. 8 and Supplementary Fig. 2). For the oceanic shallow convection, our results of low and mid-level cloud correlations consistently indicate that shallower and thinner clouds (e.g., Sc) correspond better to lighter precipitation, while higher and thicker clouds (e.g., Ns) correspond





better to heavier precipitation. In the case of Cu, the correlation coefficient is roughly the same between ocean and land for the 7th precipitation bin, but the correlation curves diverge as heavier precipitation is added. By the time the frequencies of all precipitation bins from 7th to 15th have been added, oceanic Cu clouds have twice as strong anti-correlation compared to their continental counterparts. As discussed previously in the context of Fig. 10, correlations among cloud fraction co-occurrence, i.e. [Cu vs. Cs] or [Cu vs. Cb], are not as different between ocean and land as those shown here between clouds and precipitation. The weaker anti-correlation of continental Cu cloud with rainfall reflects then, at least partly, the less robust relationship between heavy precipitation and continental high clouds.

### 3.4 Limiting factors and uncertainties

### 3.4.1 Uncertainty of cloud type classification

In this study, MODIS-observed clouds are classified into 9 cloud types adopted from previous ISCCP conventions (Chen et al., 2000; Rossow and Schiffer, 1999) for the sake of convenience. This classification is, strictly speaking, based on arbitrary $\tau$ and $p_c$ thresholds, and clouds assigned to each pair of bin boundaries will only loosely represent cloud types originally defined from morphological features seen by surface observers. Previously we noted that continental MCSs often include thick anvils (Cetrone and Houze, 2009; Yuan et al., 2011), but we can not identify whether those anvils are classified as Cs or Cb without knowledge of the cloud vertical extinction profile. Moreover, a passive sensor like MODIS has intrinsic limitations in identifying certain cloud types. Recent studies examining the nature of MODIS Cloud Regimes with active sensor observations from CloudSat and the Cloud-Aerosol Lidar and Infrared Pathfinder Satellite Observations (CALIPSO) show that similar MODIS joint histograms can have a variety of cloud vertical structures (Oreopoulos et al. 2017, submitted to *J. Geophys. Res. Atmos.*). In addition, Wang et al. (2016) showed that defining cloud types from CloudSat-CALIPSO observations where cloud vertical extent is better known can yield large disagreements with cloud type definitions from the MODIS $p_c$-$\tau$ joint histogram. Such ambiguous definitions of cloud types from passive measurements may be the reason for substantial correlations between As and certain ranges of precipitation even though As is usually thought of as a not precipitating cloud type. In summary, the 9 cloud types in this study may not strictly correspond to their traditional, ground-based classification, so their relationship with precipitation should not be taken literally or juxtaposed with empirical knowledge. They are simply a convenient framework to organize findings about cloud-precipitation covariability at 1° scales.

Furthermore, the passive MODIS observations suffer from low skill in detecting multilayer clouds. Specifically, MODIS generally only detects the cloud top of the highest cloud, so high clouds such as cirrus or stratiform anvil will mask the presence of shallow clouds. This may be a contributing factor to the negative correlations by Cu and Cs in Fig. 10. Unfortunately, this is a shortcoming of passive cloud observations that we have to contend with in exchange for wider coverage.



### 3.4.2 Uncertainty of TMPA and its temporal matching to MODIS

As noted in subsection 2.1, TMPA quality varies by location. Over land, the strong surface emissivity forces microwave retrievals of precipitation to rely on the ice scattering signature, which may not be present for warm (or shallow) rain. While there are gauge adjustments over land, they depend on the quality and density of the gauges used and operate at monthly time scales—thus may not be able to correct the precipitation rates for individual rain events. Over ocean, gauge adjustment is unavailable, leading to potential systematic errors in the precipitation estimates. Furthermore, the retrieval of remotely sensed precipitation relies on algorithms that estimate surface precipitation rates from passive microwave brightness temperature, a task that remains a challenge. In addition, due to the intermittency of passive microwave sensors on low-Earth orbit satellites, gaps in the microwave field are filled in by infrared-based precipitation estimates, which have poor accuracy as infrared brightness temperature in isolation is only indirectly related to precipitation (it is as if we were trying to correlate precipitation with one-dimensional $p_c$ histograms). Hence, TMPA estimates possess considerable uncertainties.

Furthermore, precipitating systems can develop quickly, especially over land. For example, a tropical squall line can develop in a few hours, so it is possible that MODIS and TMPA observe different stage of a system given that a 1.5 hour difference is possible despite our temporally matching. This situation can result in decreased correlation coefficients between high-thick cloud and heavy precipitation over land. We are somewhat less concerned about this sampling issue because the lead/lag time between MODIS and TMPA is expected to be random, and therefore hopefully not a source of systematic bias. In the future, this concern will be ameliorated by using a higher temporal resolution precipitation dataset such as the Integrated Multi-satellitE Retrievals for GPM (IMERG; Huffman et al., 2015) in place of TMPA.

## 4 Summary and Conclusion

The total amount, intensity, and frequency of precipitation should be organically related to the properties of the clouds from which it originates. However, due to different radiative signal strengths of hydrometeors at particular parts of the electromagnetic spectrum, precipitation and cloud observations are significantly decoupled, necessitating joint analysis of products developed for different purposes and from imperfectly matched observations. Even with such non-ideal data at hand, we still aspire to answer fundamental questions such as: To what degree can precipitation be predicted given information about clouds? Conversely, with precipitation information at hand, can we provide good guesses about the nature of the clouds responsible? Is precipitation variability associated with cloud variability? Do answers to the above questions differ substantially between ocean and land?

In order to advance the problem of understanding cloud-precipitation co-variability, we use contemporaneous multi-year datasets, widely-accepted concepts about what a cloud type is from passive observations, and a combination of compositing and correlation analysis. We try to preserve some sub-grid variability information at one-degree scales by employing precipitation histograms built from the TMPA dataset, as well as MODIS joint histograms of cloud top pressure and cloud optical thickness, both of which are matched spatiotemporally.



We find, not surprisingly, that correlations between deep convective clouds and heavy rainfall are strong and stand out clearly, dwarfing all other correlation combinations for both land and ocean. Land-ocean differences are also remarkable. For example, oceanic deep convection systems (e.g., mesoscale convective systems) are more likely to attain overcast conditions and to have larger fractions of rainy sub-cells within 1° grid cells, both indicative of larger horizontal size than their

continental counterparts, consistent with previous studies. Over land on the other hand, Cb and Cs clouds are related not only with heavy precipitation, but rather with a broader range of rainfall which translates to weaker correlations.

Thin clouds, particularly Cu clouds are anti-correlated with moderate-to-heavy precipitation. The anti-correlation is stronger over ocean, and the magnitude is comparable to the anti-correlation between Cu and high-thick clouds (Cb or Cs). The fact that oceanic deep convection often fills and outgrows the 1° reference grid cell, is ultimately the cause of clearer

relationships (less uncertainty) among heavy precipitation, high-thick clouds, and low-thin clouds.

Over ocean, low-to-mid level clouds also exhibit positive correlations with precipitation of certain ranges, which represents shallow convection and warm rain processes. Among those clouds, the relatively higher and thicker Ns clouds relate better to mid to heavy precipitation, while lower and thinner Sc clouds relate better to light precipitation. In the end, positive correlations indicate that oceanic precipitation comes from a variety of cloud types while most precipitation over land

requires the presence of high clouds. Notably, the shallow continental clouds show better anti-correlations with heavy precipitation rather than positive correlations with light precipitation. It is conceivable that this result can change once detection of low clouds in the presence of high clouds and of warm rain over land improves.

Collectively, we make a strong case that rainfall predictability is better over oceans than continents when cloud information is available. But even over oceans, there are significant uncertainties in linking certain ranges of precipitation with specific

cloud types, at least with our approach. Our self-imposed objective to make the study general, multi-year, and applicable to half of the Earth's surface, force us to resort to Level-3 gridded data, sacrificing perhaps some of the details seen in previous studies that used Level-2 data (but suffered from reduced coverage). However, our datasets are good enough to allow us to extract major features of cloud-precipitation co-variability in the tropics.

We expect that our study can form the basis of enhanced evaluation of precipitation in GCMs. A regime-based analysis in

the deep tropics by Tan et al. (2017; submitted to *Clim. Dyn.*) suggests that clouds and precipitation are more decoupled in models than in observations. Confirming that conclusion with the approach introduced in this study is a possible next endeavour. In addition, more effort should be extended to identify the optimal datasets that contain sufficient degree of detail to go along with extensive coverage in order to characterize with more confidence the cloud-rainfall relationships.

**Acknowledgements**

*We acknowledge funding from the following NASA programs: "The Science of Terra and Aqua"; "Modeling Analysis and Prediction (MAP)". JT is supported by an appointment to the NASA Postdoctoral Program at Goddard Space Flight*





*Center, administrated by USRA through a contract with NASA (NNH15CO48B). We thank our NASA colleague George Huffman for helpful discussions.*

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





## Tables

**Table 1: Population percentages of grid cells with specific precipitation characteristics over ocean and land from 13 years of data in our extended tropics domain.**

|  | Ocean | Land |
|---|---|---|
| P0>0.5 | 85.21% | 89.95% |
| P4>0 | 11.13% | 9.73% |
| P5>0 | 4.52% | 4.23% |
| P4+P5>0 | 11.41% | 10.13% |
| P4>0 and P5>0 | 4.27% | 3.83% |

5 **Figures**

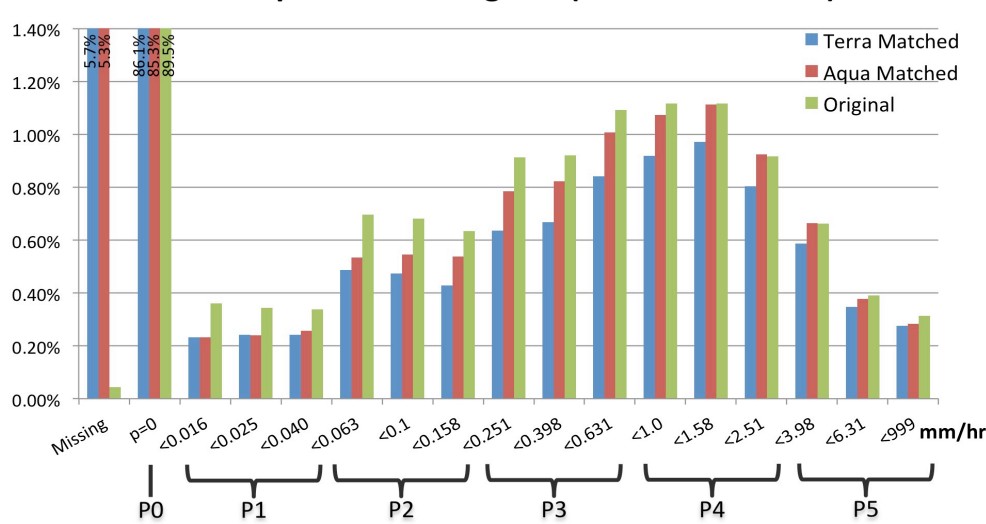

**Figure 1: Histograms of TMPA original 0.25° 3-hourly 3B42 precipitation data (green), and subsets matched with daytime Terra (blue) and Aqua (red), from December 2002 to November 2015 in extended tropics domain.**
10 **Definitions of six simplified precipitation groups are shown at the bottom.**



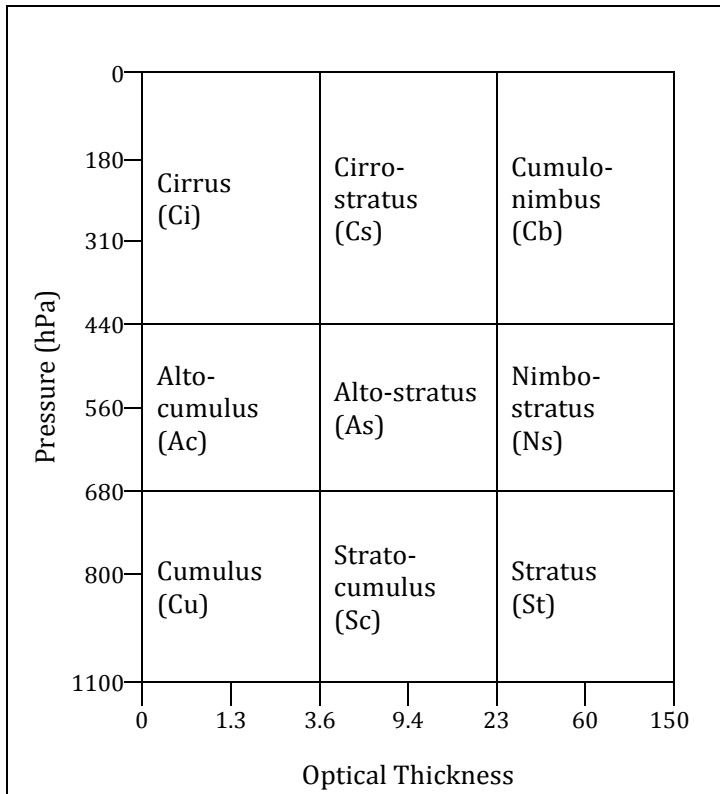

**Figure 2: ISCCP cloud types assigned to groups of bins in MODIS joint histogram of $\tau-p_c$.**



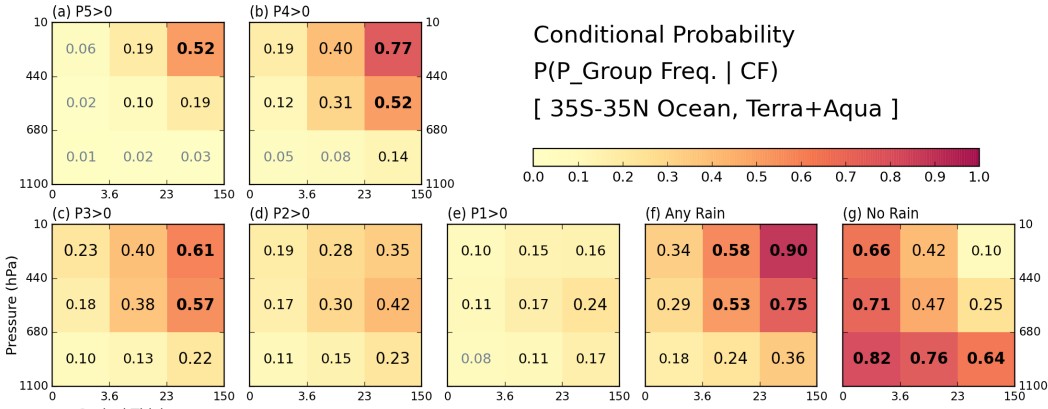

**Figure 3: (a) to (e): Conditional probabilities of precipitation within a P-group (from TMPA) given occurrences of a cloud type (from MODIS) over ocean in the extended tropics from December 2002 to November 2015; (f): Conditional probabilities of any rain amount (sum of all P-group frequencies); (g): Conditional probabilities of no rain co-occurring with cloud. The CF threshold for cloud type occurrence is 6.25%.**

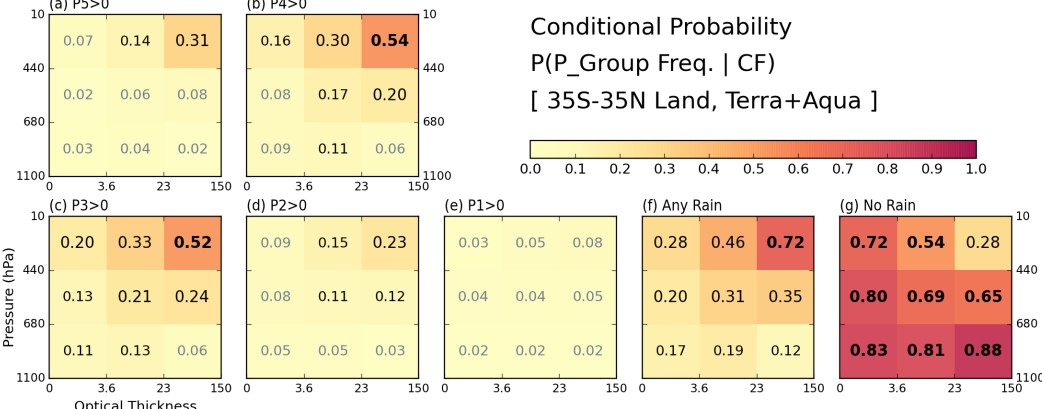

**Figure 4: Same as Fig. 3, but over land**



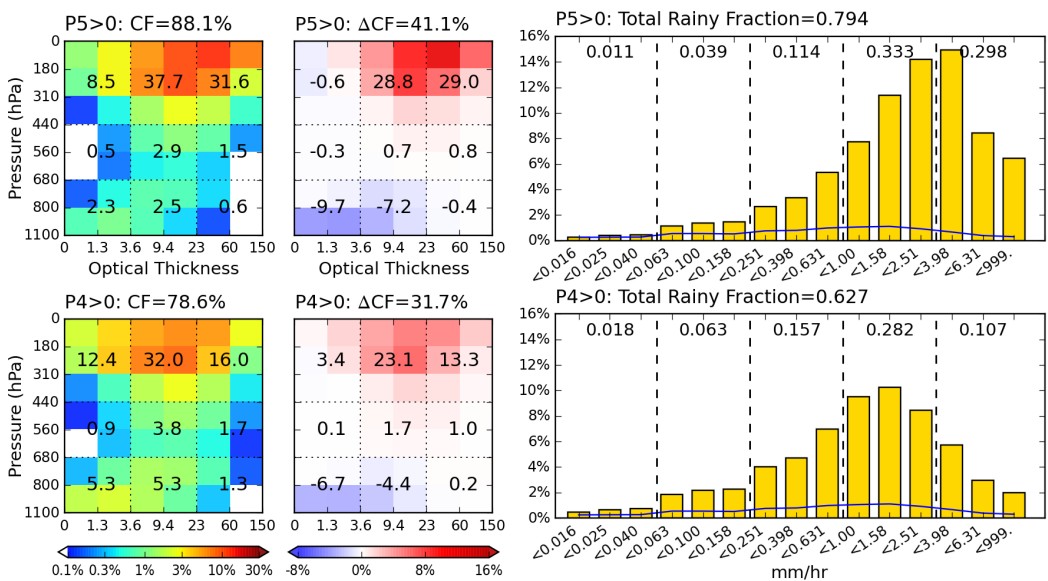

**Figure 5: Composite mean of 2D joint histogram of $p_c$ and $\tau$ (left column), differences from climatology (middle column) and precipitation histogram (right column) over the extended tropical oceans for 13 years. Top row is for P5, while bottom row is for P4 precipitation. Blue lines in precipitation histograms indicate climatology. Both cloud and precipitation climatologies correspond to the entire domain, not just for ocean. Numbers on cloud histograms are the cloud fraction (CF; %) of each cloud type, which is the sum of 4 or 6 histogram bin values assigned to the cloud type. The sum of all values is equal to the total cloud fraction shown in panel titles. Numbers on precipitation histograms are the fraction of each P-group, P1 (left) to P5 (right), which is the sum of three individual bin values. Total rainy fraction is the sum of all P-groups' fractions (i.e., sum of 15 individual bin values).**

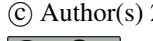



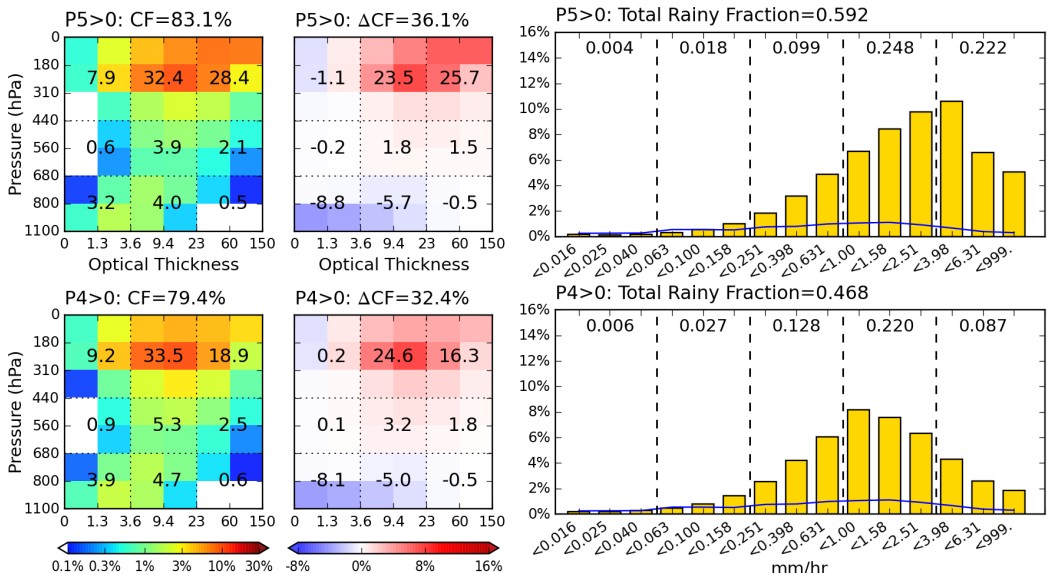

Figure 6: Same as Fig. 5, but over land.



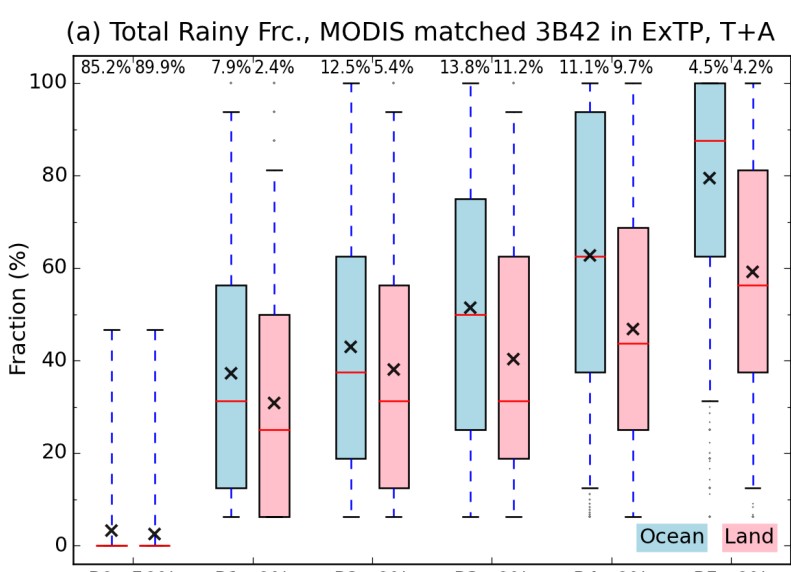

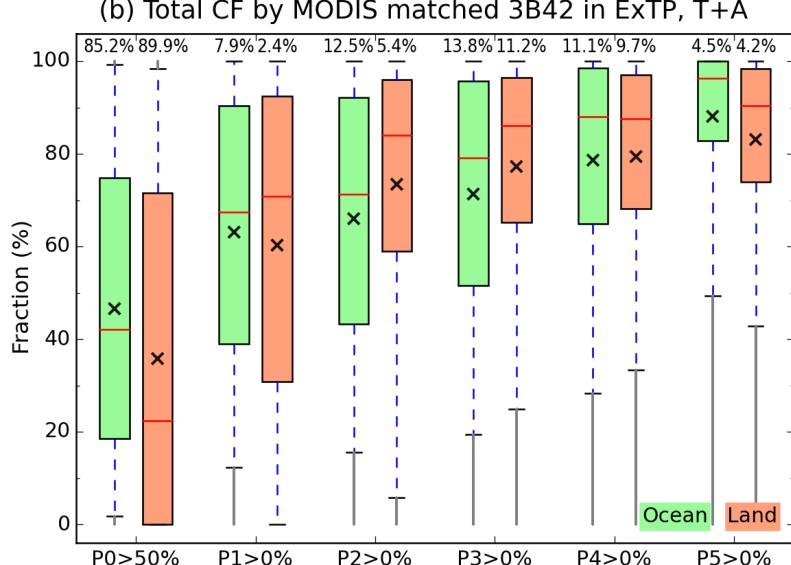





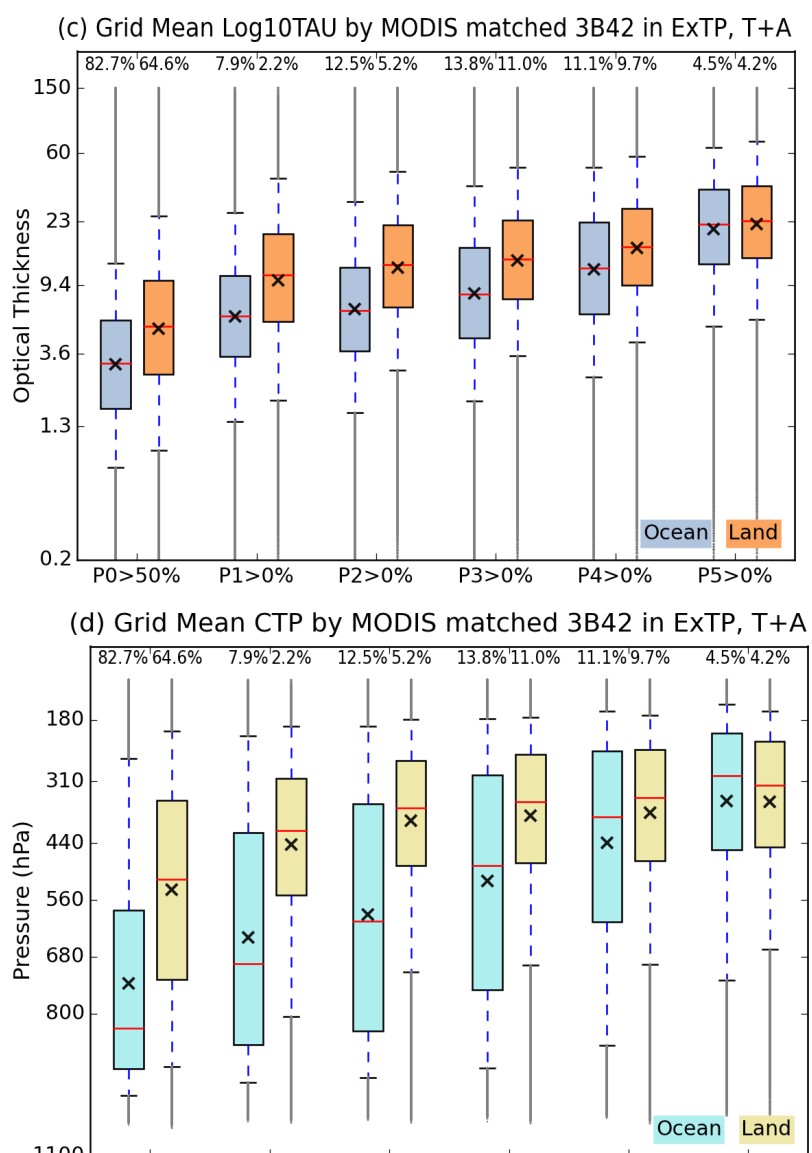

**Figure 7: Box-whisker plot of (a) the total rainy fraction, (b) the total cloud fraction, (c) the grid mean *log10(τ)*, and (d) the grid mean *pc* conditioned by precipitation groups, separately for ocean and land. The median values are shown as red horizontal lines, and the mean values are shown as black crosses. The vertical width of the boxes indicates the**





interquartile range (25$^{th}$-75$^{th}$ percentile), and the whiskers extend from 5% to 95% values. Percentage numbers on top of the boxes indicate the occurrence ratio of each P-group relative to the total ocean or land grid cells.

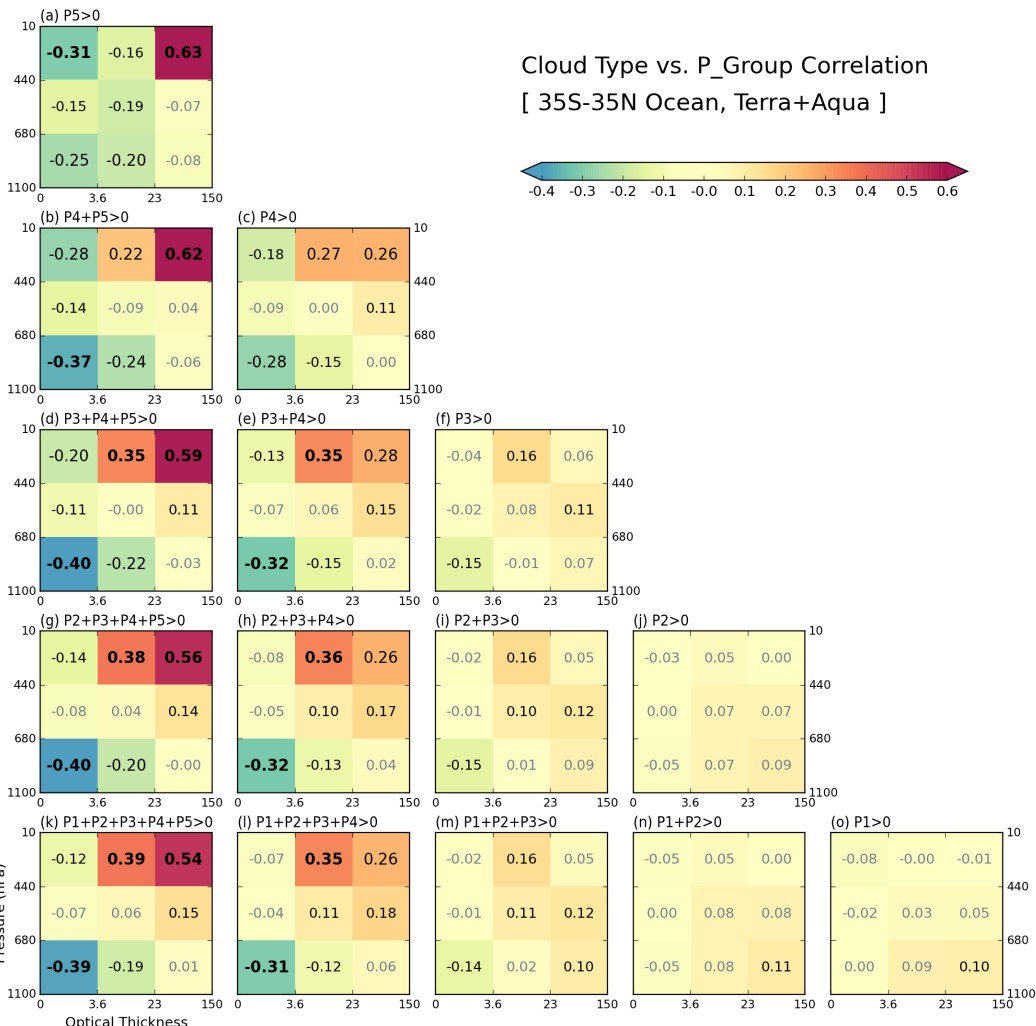

**Figure 8: Cross-correlation coefficients in the extended tropical oceans for 13 years calculated between CFs of cloud types and precipitation group (individual or cumulative P-groups) values. The sum of all five precipitation groups shown in (k) indicates the total rainy fraction.**



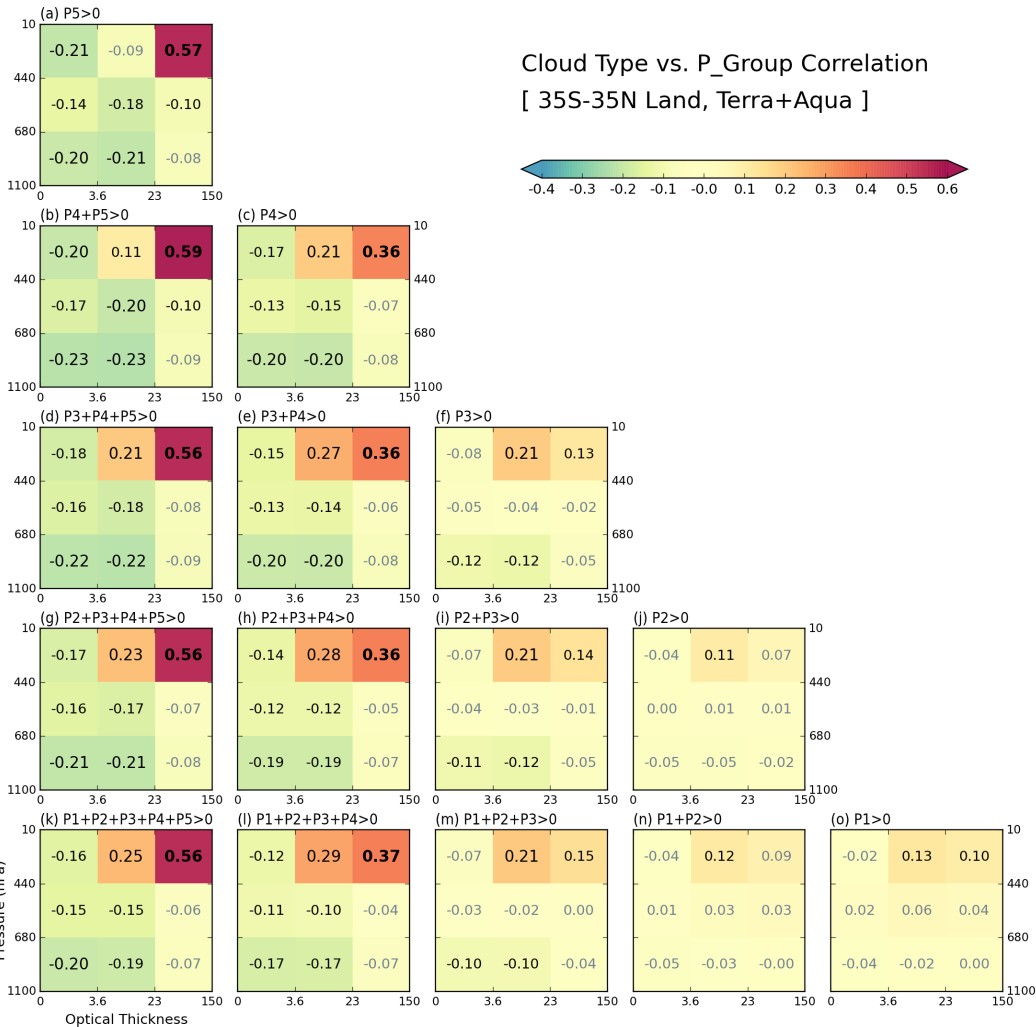

**Figure 9:** Same as Fig. 8, but over land.





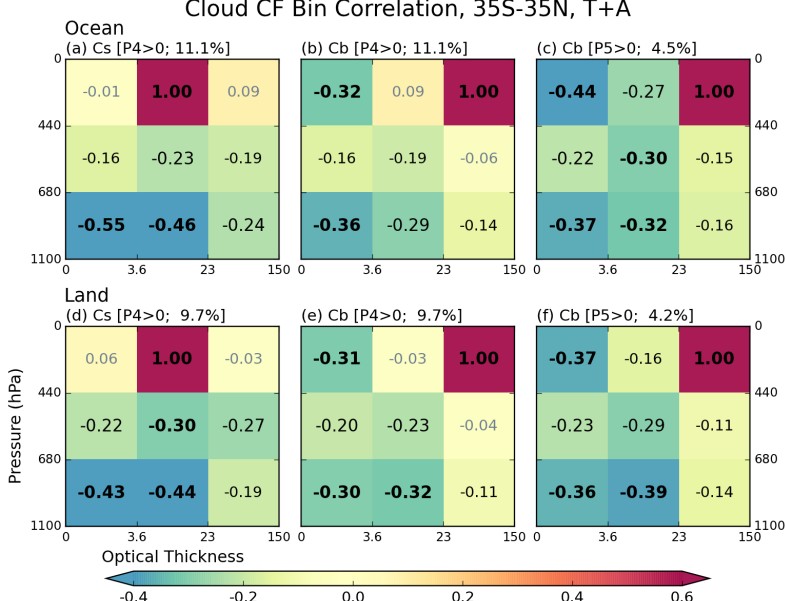

**Figure 10. Cross-correlation coefficients are calculated between cloud joint histogram bin CF values for 13 years, based on (a) Cs CF over Ocean when P4>0, (b) Cb CF over Ocean when P4>0, (c) Cb CF over Ocean when P5>0, (d) Cs CF over Land when P4>0, (e) Cb CF over Land when P4>0, and (f) Cb CF over Land when P5>0. Percentage of subtitle indicates sample size ratio over total ocean or land grid cells.**

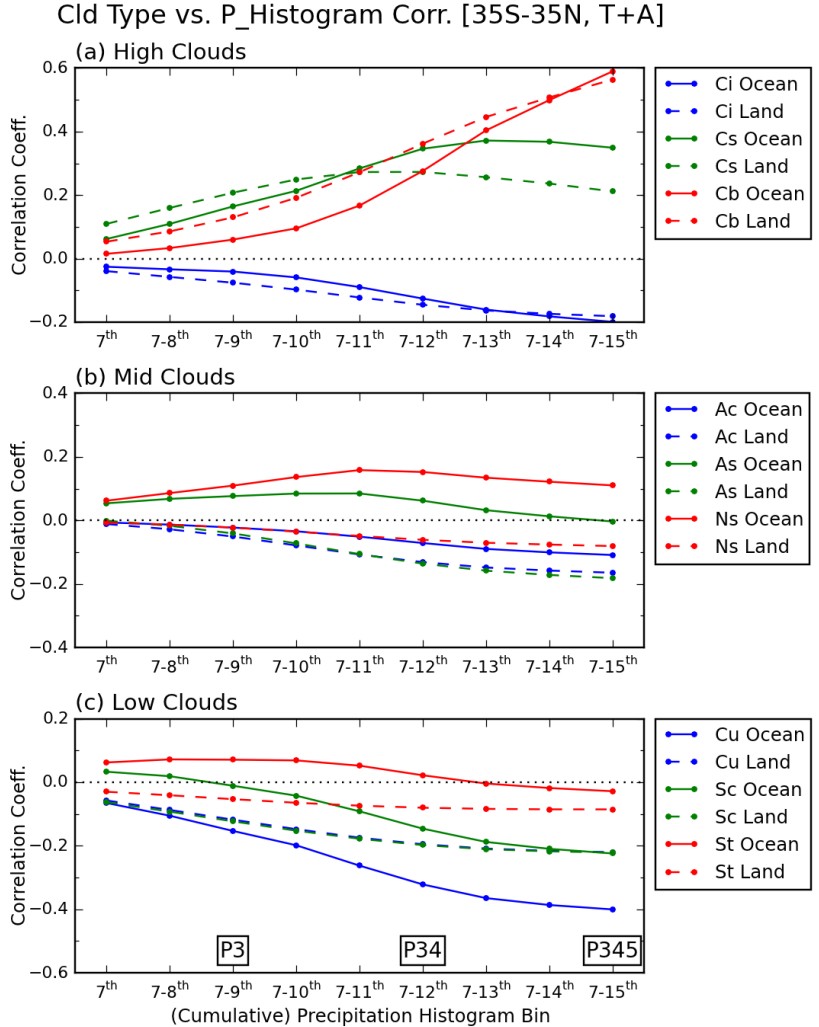

**Figure 11: Correlation coefficients between cloud type CF and precipitation histogram values, for (a) high clouds (Ci, Cs, and Cb), (b) Mid-level clouds (Ac, As, and Cu), and (c) low clouds (Cu, Sc, and St). Precipitation histogram values are accumulated from the 7th bin onward, so the sum from the 7th to the 9th bin is the same P3, and so on. Ocean clouds are shown in solid and land clouds are shown in dashed lines.**