# Peer review of "Contrasting the Co-variability of Daytime Cloud and Precipitation over Tropical Land and Ocean"

_Atmospheric Chemistry and Physics, 2017_

## Referee Comment (RC1) · Anonymous Referee #1 · 14 Sep 2017

General comments:

Accurate knowledge of the relationship between clouds and precipitation is a key aspect for climate and earth-system models. Hence, many research studies previously addressed this issue by exploring cloud-precipitation feedbacks using, e.g. numerical and synoptic approaches, on various scales. The manuscript acp-2017-612 revisits this topic and aims at improving the present understanding of daytime cloud-precipitation co-variability by looking at collocated cloud and precipitation observations over a very large spatial domain, i.e. tropical oceans and land, with improved spatial and temporal matching compared to previous studies. The presented method is well suited for analysing this particular coupling and the authors provide an exhaustive analysis based on available data advancing the knowledge on this topic. Many interesting questions

are addressed and the authors draw plausible conclusions such as the stronger positive correlation between cumulonimbus clouds and heavy precipitation over oceans as opposed to over land. The manuscript is generally well written and the results are presented in a good manner. The manuscript fits well within the scope of Atmospheric Chemistry and Physics and I recommend publishing it after the authors have corrected several minor issues. In general, the manuscript contains heaps of detail but it is unfortunately kept very descriptive, and generally lacks more conclusions and practical implications of the presented findings. In addition, it would be highly beneficial if the authors stress the importance of their results and explicitly state what new scientific insights have been discovered by means of their analysis and which previous knowledge could be confirmed / or rejected. The conclusion section should be suitable for this. A critical examination of shortcomings of the method and data sets is quite brief and it is not entirely clear to what extent the individual sensitivity of the chosen data products for clouds and precipitation may bias the results. In specific, would the authors come to similar conclusions with different data products?

Specific comments:

P2 L26: Please provide the motivation for your study already here and explain in more detail what is missing in previous research studies. At this point it is not clear to the reader why this topic needs to be touched 'once again'.

P2 L32: Move this part up accordingly.

P3 L1: What is meant by 'ambiguity' exactly?

P3: Please elaborate in detail why the MODIS and TRMM TMPA were chosen. It would be highly beneficial to discuss and argue why these two data sets are more suitable than other similar data sets for your study. What about other global precipitation products providing 3-hourly rain estimates such as CMORPH (Joyce et al., 2004), PERSIANN (Sorooshian et al., 2000) or others? The sensitivity to precipitation and different cloud types may be very different amongst these products which could potentially strongly affect your findings.

P4 L24: Please be consistent in naming and differentiating between the grid at 1°
resolution and a lower resolution grid at 0.25° resolution throughout the paper (grid
cell, sub-grid cell, sub-grid, sub-cell, etc.)

P5 L3-11: The spatial and temporal collocation is crucial for this type of study as pre-
cipitation and rain patterns may vary quickly. Please provide a better explanation of the
temporal matching of both data sets and provide a reference for the time conversion
of the MODIS data, if possible. The TRMM TMPA 3B42 3-hourly product provides the
satellite observation time for each grid cell. Was this information used for the matching?
Note that actual observation times for each grid pixel may vary +-90 minutes within a
3-hourly data file. For example, if the TMPA 3B42 12UTC data file is chosen for collo-
cation with MODIS Aqua data the maximum time difference between the MODIS and
TMPA data could be more than 1.5 hours. It is generally not quite clear to the reader
how the non-trivial collocation of the data sets is performed across all longitudes.

P5 L19-21: What could be the explanation for this?

P6 L7-9: Why did you choose to consider the MODIS Terra and Aqua as a single
ensemble, even though initial results from Fig. 1 point at notable differences in precipi-
tation during the different overpass times? Would you argue that this has no effect on
the found cloud-precipitation relationships? Also, it is not clear to the reader how the
exact matching of MODIS and TRMM TMPA data is performed. See remark above.

P7 L32-33: Please explain why this is the case for P5 and not for P4 and name the
common characteristics with MCS explicitly.

P9 L11: It would be worthwhile to explain the effect of autocorrelation between neigh-
boring grid cells in more detail and how this is accounted for.

P9: How certain are the authors that the calculated correlation coefficients between
the cloud types and precipitation data can be interpreted as a 'general relationship'

and not just representing the sensitivity of the TRMM TMPA algorithm to different cloud types?

P13 L2: It would have been quite interesting to see how the correlation coefficients for Fig. 11 change if precipitation frequencies are not progressively added for each bin. Could you provide such a Figure or give a reason why this may not be useful?

P16 L27-28: This sentence sounds a bit strange, suggesting that you might not have chosen the optimal data sets for your study in the first place. It would be better to discuss in more detail how your results could be validated against results derived from other or future data products.

Technical corrections:

P1 L22-23: Please rephrase to make points clearer

P1 L30: 'models' instead of 'model'; or better rephrase the first part of the sentence

P2 L18: Please rephrase very long sentence

P2 L20: 'larger extent' - please state whether horizontal or vertical extent is meant

P2 L26: Please specify references to the datasets used

P2 L33: 'examines' instead of 'examine'

P3 L3: 'Is there a closer' instead of 'Is there more close'

P3 L11: Please provide references for the MODIS instrument and specify the exact name of the dataset.

P3 L22: Please rephrase and/or explain why a lower number of bins was chosen.

P3 L26: leave out 'best'

P4 L1: Specify the overpass time of MODIS Terra and Aqua in local time / equator crossing time.

P4 L14: Please rephrase part with 'algorithmic variations'

P4 L19: Please rephrase the first part of the sentence

P4 L21-22: Use consistent format of grid resolution, i.e. either 1° or 1x1°

P4 L24: 'of a histogram', instead of 'of histogram'

P4 L24: 'without missing values' instead of 'when no missing values exist'

P4 L32: 'shows the distribution' instead of 'shows distribution'

P5 L3: Please rephrase the sentence, for example: '... the TMPA and MODIS observations also need to be matched in time.'

P5 L15: Please rephrase 'in explaining'

P5 L16: Please rephrase 'For example' at the beginning of the sentence

P5 L17: What is meant exactly by: 'relatively suppressed'?

P5 L28: Please rephrase sentence starting with "This is simply . . ." as this would have been probably possible and could, in fact, provide additional insight, but was not pursued for practical reasons.

P6 L4-7: Please rephrase last part of sentence starting at "with no confusion resulting"

P6 L13: Please explain what it is meant by the co-variability of anomalies.

P6 L27: Replace 'of no-rain case' with 'of the no-rain case'

P7 L3: Please indicate in which section the issue of less rain over land is analysed.

P7 L4: Please make it clearer to reader what you mean by the 'composite mean cloud and precipitation histogram'

P7 L19: 'in the P4 group' instead of 'due to the P4 group'

P8 L13 and L16: is the increase really linear?

[Figure]

P8 L29: Please rephrase 'a factor affecting the Fig. 7 results"

P10 L6: Please rephrase 'the peak negative value is a weaker value of'

P10 L24: Please rephrase the second part of the sentence to make clear what you mean exactly.

P12 L8: Please rephrase 'but suffice it to say here'

P13 L11: Please rephrase the beginning of the sentence (not start with 'But')

---

## Referee Comment (RC2) · Anonymous Referee #2 · 13 Nov 2017

In "Contrasting the Co-variability of Daytime Cloud and Precipitation over Tropical Land and Ocean", Jin et al. analyze satellite datasets of cloud and precipitation retrievals with the aim of determining the relationship between cloud and rain.

The topic is highly relevant for understanding the behavior of the atmosphere, both in physical reality and in parameterized cloud and precipitation in models. To my knowledge, the authors are the first to use this particular technique of regime-based cloud classification to analyze the relationship between cloud and precipitation, and I therefore recommend the analysis be published.

The authors have chosen a fairly non-straightforward analysis method, and I hope the comments below will help them clarify a few points for the reader.

[Figure]

The main potential weaknesses of the analysis are the following:

1. As the authors themselves point out, using cloud optical thickness and cloud-top pressure to define cloud regimes is an essentially ad-hoc classification based on arbitrary choices. In the conclusions, they then describe the regime classification as "widely accepted". It is true that these regimes are widely used, subject to the known caveats that the authors correctly state (e.g., that the regime names are not to be taken to correspond literally to actual cloud types); however, this acceptance is based on the regimes' usefulness having been demonstrated for each particular application, for example by showing that susceptibilites to aerosol are very different across regimes (e.g., Gryspeerdt et al, ACP 2014). In my opinion, this paper provides some interesting indications that the regime classification does indeed differentiate between cloud type of very different behavior regarding precipitation, but this is the case for only three (out of nine) regimes, so I think it will take some extra work (perhaps the unpublished paper referred to in the Conclusions) before the field will "widely accept" the use of these regimes for precipitation studies.

2. Many of the conclusions are based on regime-composite Pearson correlation coefficients between cloud area fraction and precipitation intensity percentiles. The Pearson correlation coefficient is fraught with pitfalls. The authors would greatly assist the reader in his or her assessment of the robustness of the conclusions by providing:

   (a) a representative scatter plot of the correlated variables in the case of a strong positive correlation and a strong negative correlation and

   (b) a geographic map of correlation strengths for the strongly positively and negatively correlated cloud/precip categories to see, e.g., whether the subsidence regions, ITCZ, warm pool, SPCZ, and maritime continent contribute as expected to the global-mean positive and negative correlations.

3. According to the authors, the TMPA precipitation dataset uses cloud-top temperature to fill in precipitation information where radar is not available. Since cloud-top height information is also used in the regime definitions, I would expect some amount of potentially spurious correlation. Discussion of whether this effect has been considered would be appropriate in the text.

Minor comments:

- Section 2.2, "If the number of bins in the histogram is chosen to be also 16, each bin value falls between 0 and 1 in multiples of 1/16, the sum of all histogram bins at $1°$ grid cell is equal to 1, and sub-grid precipitation rates are thus converted to areal fractions of specific ranges of precipitation rates": I don't quite follow the why 16 is a magic number in the link between the number of bins and area fraction; since we end up with 6, not 16, bins in Sec. 2.3, are those not area fractions anymore?

- Fig. 5: define what is meant by "climatology".

- p. 11, first paragraph: I find the claimed link between P4, P5, and MCS tenuous; for example, if P5 indicates MCS (where we expect clouds at all levels), why are both Cb and Cs anticorrelated with low- and mid-level clouds?

- Anticorrelation in the Cu case: I am surprised that Cu is so anticorrelated with rain; I always thought (perhaps my thinking is guided by the regime name, which the authors caution against) that this would be the regime that clouds with high in-cloud water content but low area fraction (hence low grid-scale optical thickness).

- Anticorrelation in the Cu case (still): It would be interesting to get to the bottom of whether this is a real effect (CAPE/stability) or shadowing artifact, and I think the authors could easily do it by looking at CloudSat profiles (since they are already

using MODIS data, not much additional co-location would be needed). If it is an artifact, does that mean all of Fig. 8 could be simplified to just the first row of every 3×3 plot? (By the way, I think the matrix of additive/subtractive $P_n > 0$ subsets in Figs. 8 and 9 is brilliant plotting strategy.) Anyway, my first guess at the source of the anticorrelation was open vs closed-cell stratocumulus, and it was interesting to learn that that was not the reason.

- The other surprise for me in Fig. 8 is that cor(CF, $f_{prec}$) never goes above 0.6. All CBs precipitate, so I would expect the Cb CF should correlate much more strongly with $f_{prec}$. What am I missing?

- Sec. 4: it should be clarified that the first paragraph is an aspirational statement about the cloud-physics field as a whole, since this study is an incremental advance

- p.16 l.16: if "once detection of low clouds in the presence of high clouds and of warm rain over land improves" refers to the use of active rather than passive satellite sensors, the authors may be interested in Field and Heymsfield or Mulmenstadt et al (both 2015, GRL)

- p.16 l.20: The authors chose not to use L3 instead of L2 data, presumably for reasons of data management complexity. I don't think anyone would fault them for this choice, so the defensive tone of this sentence is out of place. Either that, or I misunderstood something about it.

- p. 16 l.25: No objection to citing unpublished work, but why not also some published references that show the same thing, e.g., Suzuki et al (2015, J Atmos Sci), Jing et al (2017, JGR)

---

## Author Comment (AC1) · 21 Dec 2017

**General comments:**

*Accurate knowledge of the relationship between clouds and precipitation is a key aspect for climate and earth-system models. Hence, many research studies previously addressed this issue by exploring cloud-precipitation feedbacks using, e.g. numerical and synoptic approaches, on various scales. The manuscript acp-2017-612 revisits this topic and aims at improving the present understanding of daytime cloud-precipitation co-variability by looking at collocated cloud and precipitation observations over a very large spatial domain, i.e. tropical oceans and land, with improved spatial and temporal matching compared to previous studies. The presented method is well suited for analysing this particular coupling and the authors provide an exhaustive analysis based on available data advancing the knowledge on this topic. Many interesting question are addressed and the authors draw plausible conclusions such as the stronger positive correlation between cumulonimbus clouds and heavy precipitation over oceans as opposed to over land. The manuscript is generally well written and the results are presented in a good manner. The manuscript fits well within the scope of Atmospheric Chemistry and Physics and I recommend publishing it after the authors have corrected several minor issues. In general, the manuscript contains heaps of detail but it is unfortunately kept very descriptive, and generally lacks more conclusions and practical implications of the presented findings. In addition, it would be highly beneficial if the authors stress the importance of their results and explicitly state what new scientific insights have been discovered by means of their analysis and which previous knowledge could be confirmed / or rejected. The conclusion section should be suitable for this. A critical examination of shortcomings of the method and data sets is quite brief and it is not entirely clear to what extent the individual sensitivity of the chosen data products for clouds and precipitation may bias the results. In specific, would the authors come to similar conclusions with different data products?*

We thank the reviewer for the overall positive assessment. At the outset of the study we aimed to answer specific questions (P3 L3-6) and we believe we generated sufficient results to shed light on these issues. However, we may have not always provided direct answers, something we're trying to improve upon in the revised version. We think that the value of this study rests in employing a novel methodology where cloud histograms and matched precip histograms are jointly analyzed to re-examine in an extensive

semi-global-domain results previously reported in local/regional studies. We quote many specific results in the concluding section, where every cloud type's correlation with precipitation is summarized. We also consider the conclusion that shallow continental clouds are better anti-correlated with heavy precipitation than correlated with light precipitation very specific and useful from the perspective of current precip and cloud measurement capabilities and our ability to predict precip from cloud information. Following the suggestion of the reviewer, the higher-level summary has been refined as well ("While some … this conclusion"). Regarding the biases of the specific datasets, explanations are provided below in responses to specific comments.

(P17 L8) "While some of the details seen in previous studies that used Level-2 data will unavoidably be lost, our datasets are good enough to extract major features of cloud-precipitation co-variability and allow us to claim that they are broadly representative of this co-variability in the tropics. We argue that the insensitivity of cloud-precipitation relationships to location (supplementary Fig. 4) and precipitation dataset (initial tests with recent GPM-IMERG data that may be presented in a future study yielded similar results) strengthen the validity of this conclusion."

**Specific comments:**

*P2 L26: Please provide the motivation for your study already here and explain in more detail what is missing in previous research studies. At this point it is not clear to the reader why this topic needs to be touched 'once again'.*

Motivation was provided in the paragraph from P2 L26 to P3 L7. See also response to related question below.

*P2 L32: Move this part up accordingly.*

To address the above comment, the paragraph has been re-written:

(P2 L28) "We thus strive for generality of results by covering the entire tropics and for overcoming the ambiguity of CR-based studies by taking advantage of the ability to break down individual grid-box cloud fractions with the aid of joint cloud histograms. Hence, our paper revisits and explores anew the mesoscale cloud-precipitation relationship via the synoptic approach by employing a Moderate Resolution Imaging Spectroradiometer (MODIS) gridded cloud dataset (King et al., 2003; Platnick et al. 2003) and the TRMM Multi-satellite Precipitation Analysis (TMPA) dataset (Huffman et al., 2007, 2010). While the MODIS Level-3 data are provided at 1°×1° resolution, the 2D joint histogram of cloud optical thickness ($\tau$) and cloud top pressure ($p_c$) contains pixel-level cloud information which can be combined with the sub-grid variability of precipitation at the 1°×1° scale, available by virtue of the finer 0.25°×0.25° spatial resolution of TMPA. While still coarser than the TRMM PR dataset, the combined MODIS and

TMPA dataset covers the entire tropics every single day, allowing better generalization of the daytime relationship between clouds and precipitation. We seek to answer questions such as: …"

*P3 L1: What is meant by 'ambiguity' exactly?*

This is in reference to lines 13-14 of p. 2 where it is noted that the CR internal variability can be quite large.

*P3: Please elaborate in detail why the MODIS and TRMM TMPA were chosen. It would be highly beneficial to discuss and argue why these two data sets are more suitable than other similar data sets for your study. What about other global precipitation products providing 3-hourly rain estimates such as CMORPH (Joyce et al., 2004), PERSIANN (Sorooshian et al., 2000) or others? The sensitivity to precipitation and different cloud types may be very different amongst these products which could potentially strongly affect your findings.*

The main reason for choosing MODIS and TMPA was our more intimate knowledge of these datasets. Moreover, our working assumption was that in such well-characterized Level-3 datasets, severe biases (if they do exist) are limited or well-known. On the precipitation side, we applied our methodology to GPM-IMERG and CMORPH for three recent years, and found that the correlation pyramid plots are quite similar to the one shown in this study. Only minor differences in the coefficient numbers were seen, and different peak locations for light precipitations, particularly with GPM-IMERG (as expected, since TMPA exhibits a weakness in light rainfalls as pointed out in the manuscript). However, at least for the P3, P4, P5 precipitation ranges, we didn't find any fundamental deviations from our current TMPA precipitation dataset. A sentence is added to the concluding section related to this issue. With regards to clouds, MODIS is considered the state-of-the-art provider of Level-3 1° gridded cloud histograms. Had we used ISCCP, we would have resorted to a coarser resolution of 280km and smaller overlapping period with TMPA.

(P17 L11) "We argue that the insensitivity of cloud-precipitation relationships to location (supplementary Fig. 4) and precipitation dataset (initial tests with recent GPM-IMERG data that may be presented in a future study yielded similar results) strengthen the validity of this conclusion."

*P4 L24: Please be consistent in naming and differentiating between the grid at 1° resolution and a lower resolution grid at 0.25° resolution throughout the paper (grid cell, sub-grid cell, sub-grid, sub-cell, etc.)*

Thanks for pointing this out. We now use consistently "1°×1°" and "sub-grid."

*P5 L3-11: The spatial and temporal collocation is crucial for this type of study as precipitation and rain patterns may vary quickly. Please provide a better explanation of the temporal matching of both data sets and provide a reference for the time*

*conversion of the MODIS data, if possible. The TRMM TMPA 3B42 3-hourly product provides the satellite observation time for each grid cell. Was this information used for the matching? Note that actual observation times for each grid pixel may vary +-90 minutes within a 3-hourly data file. For example, if the TMPA 3B42 12UTC data file is chosen for collocation with MODIS Aqua data the maximum time difference between the MODIS and TMPA data could be more than 1.5 hours. It is generally not quite clear to the reader how the non-trivial collocation of the data sets is performed across all longitudes.*

We are basically matching MODIS UTC time to the closest UTC time in the TMPA dataset. As a first step, we need to calculate an approximate UTC time for MODIS since it is not provided in the L3 data. We accomplish this using the grid-mean solar zenith angle $\theta_0$ and the following equation:

$$h = \cos^{-1}\left(\frac{\cos\theta_0 - \sin\varphi\sin\delta}{\cos\varphi\cos\delta}\right)$$

where $h$ is the hour angle, $\varphi$ is latitude, and $\delta$ is solar inclination angle. [Reference: Liou, K. N.: An introduction to atmospheric radiation, 2. ed., International Geophysics Series. Vol 84, Acad. Press, San Diego, 2002]. The solar inclination angle for a particular latitude is a function of the Julian date. Once the hour angle ($h$) is obtained it is converted to UTC. With the MODIS UTC at hand, we search for the TMPA 3-hour interval that contains it. For example, a TMPA data point designated as 12pm UTC contains rainfall observations from 10:30am UTC to 1:30 pm UTC, and is selected for that 1°×1° grid cell if the MODIS UTC also falls within that time interval. Hence, there is indeed possibility for a maximum time difference greater than 1.5 hours. However, we don't think that this fact can affect our results significantly because: 1) The distribution of satellite observation times within a 1°×1° grid cell and a 3-hour interval is probably closer to a Gaussian distribution rather than Uniform distribution because, when two or more satellite rainfall observations are available, the one closest to the middle of 3-hour period is selected by the algorithm, and 2) Time mismatch biases are random and largely cancel out due to huge sample size. The text in p. 5 has been changed accordingly to further elaborate on these issues, and a statement about maximum time difference has been removed.

(P5 L11) "The UTC of each grid cell can be estimated from the mean solar zenith angle (SZA) available as a MODIS Level-3 variable, and the latitude and time information for each grid cell."

(P5 L15) "Since the TMPA data is available at 3 hour-intervals, TMPA data centered, say, at 12 pm UTC, will be matched with MODIS data having UTC between 10:30 am and 1:30pm."

*P5 L19-21: What could be the explanation for this?*

For comparing Aqua matched precipitation against all available precipitation, differences in missing is 5.3% (Aqua: 5.34%, All: 0.04%), and in "No Rain" is -4.19% (Aqua: 85.28%, All: 89.47%). Hence, 1.11% of All data (5.3%-4.19%) is distributed to various precipitation histogram bins, 0.07% per bin on average, thus it is normal that All available data (green bar) is slightly higher than Aqua-matched data (red bar).

For the weak-to-moderate precipitation rate, the difference is slightly larger despite the intrinsic difference explained above. We interpret this as light rain being more frequent outside the time window around noon. We have rephrased the relevant passage ("This appears in Fig. 1 … overpasses") in order to reduce any possible confusion.

(P5 L24) "This appears in Fig. 1 as Terra-matched precipitation having smaller frequencies than the original and the Aqua-matched precipitation, although it is somewhat improper to directly compare Terra- or Aqua-matched data with fully sampled data because the higher ratio of available (non-missing) data in the fully sampled data propagates as higher relative frequency in the various precipitation bins. It is also notable that, for weak-to-moderate precipitation rate (less than 1mm/hr), even Aqua-matched precipitation is (slightly) lower in percentage terms than fully-sampled TMPA precipitation, which can be interpreted as weak-to-moderate precipitation being more frequent outside the time windows of Terra and Aqua overpasses."

*P6 L7-9: Why did you choose to consider the MODIS Terra and Aqua as a single ensemble, even though initial results from Fig. 1 point at notable differences in precipitation during the different overpass times? Would you argue that this has no effect on the found cloud-precipitation relationships? Also, it is not clear to the reader how the exact matching of MODIS and TRMM TMPA data is performed. See remark above.*

The Terra- and Aqua-matched precipitation does indeed look slightly different in terms of the sample distribution of Fig. 1. At the same time, the cloud type distribution should also not be exactly the same between Terra and Aqua considering the diurnal cycle of cloudiness in some locations (especially continents). However, in this study, we are looking for a general relationship between clouds and precipitation that signifies physical processes (that may be different between ocean and land), and we therefore assume that this relationship is NOT affected by the frequency of specific types of cloud or precipitation. This is the main reason discrimination between Terra and Aqua data was not considered a priority in this study. We actually verified behind the scenes that separate correlations for Aqua and Terra are not too different. A second reason is that we are already breaking down the results by land/ocean, and an additional decomposition would make the presentation cumbersome and hard to follow.

*P7 L32-33: Please explain why this is the case for P5 and not for P4 and name the common characteristics with MCS explicitly.*

In the manuscript, we interpreted MCS as the combined system of strong stratiform precipitation paired with Cb clouds. In the P4 and P5 PC-TAU composites of Figs. 5 and 6, the P4 composite looks more closely related to Cs than Cb, while the P5 composite shows as many above average Cb clouds as Cs

clouds. This enhanced Cb CF is the reason why we relate P5 to MCS and not P4. The sentence has been rephrased to clarify this.

(P8 L12) "The P5 composite patterns of cloud and precipitation shown in Fig. 5 are in accordance with such MCS characteristics, i.e. strong convective clouds and a broad spectrum of precipitation."

*P9 L11: It would be worthwhile to explain the effect of autocorrelation between neighboring grid cells in more detail and how this is accounted for.*

We have added a sentence ("Consideration for … underestimated") explaining that when not accounting for the autocorrelation the degrees of freedom are overestimated and thus the significance level underestimated. In other words, because of fewer independent measurements it is harder to surpass the threshold of significance. We refer to the citations for explanation since we don't think a digression is appropriate in this case.

(P9 L23) "Consideration for the effect of neighboring grid cells is important because neighboring grid cells are usually *not* independent (e.g., a cloud system can occupy multiple grid cells); without this consideration, the degree of freedom will be overestimated, and thus the significance level underestimated."

*P9: How certain are the authors that the calculated correlation coefficients between the cloud types and precipitation data can be interpreted as a 'general relationship' and not just representing the sensitivity of the TRMM TMPA algorithm to different cloud types?*

As stated in response to a previous comment, we repeated our analysis with GPM-IMERG data and CMORPH, and didn't find any fundamental deviations from the results shown here, which supports the "general relationship" interpretation.

*P13 L2: It would have been quite interesting to see how the correlation coefficients for Fig. 11 change if precipitation frequencies are not progressively added for each bin. Could you provide such a Figure or give a reason why this may not be useful?*

Response:

This information is actually plotted, albeit at coarse binning: it exists in the P1>0, … P5 >0 (rightmost of each row) panels of the pyramid plots of Figs. 8 and 9. The more detailed graph that the reviewer suggests can now be found in the supplementary Fig. 6. It is the same format as Fig. 11, but the x-axis starting at the 1$^{st}$-3$^{rd}$ bins, running sum of three consecutive histogram bins. The plot looks consistent to Fig. 11, and with the other results of our study.

*P16 L27-28: This sentence sounds a bit strange, suggesting that you might not have chosen the optimal data sets for your study in the first place. It would be better to discuss in more detail how your results could be validated against results derived from other or future data products.*

This study provides a methodology of how to quantify the cloud-precipitation co-variability, and optimal datasets can vary depending on the purpose of a study (e.g., high resolution data would be need for a regional/seasonal study). Optimal data may not even exist now, but may become available in the future, e.g., cloud and precipitation observations of higher sensitivity from the same observational platform which would eliminate a lot of the uncertainty of inexact spatiotemporal matching. This sentence also alludes to the possibility that a higher spatial and temporal resolution precipitation dataset such as GPM-IMERG could be used for this type of analysis once the period of availability has extended substantially. The last sentence of the text has been rephrased to convey what we had in mind more clearly.

(P17 L17) "In addition, more effort should be extended to apply the framework in this study to various case studies with more appropriate datasets (e.g., using higher resolution precipitation dataset for regional/seasonal studies, or longer period dataset for climate studies) in order to increase further our degree of confidence about the cloud-rainfall relationships."

***Technical corrections:***

*P1 L22-23: Please rephrase to make points clearer*

We have rephrased as follows:

(P1 L22) "Weak correlations between weaker rainfall and clouds indicate poor predictability for precipitation when cloud types are known, and this is even more true over land than over ocean."

*P1 L30: 'models' instead of 'model'; or better rephrase the first part of the sentence*

We think the sentence reads fine after changing "model" and "AGCM" to plural.

*P2 L18: Please rephrase very long sentence*

We're not sure which sentence the reviewer is referring to here. None in the paragraph starting in L15 seem overly long. Nevertheless, we broke the sentence starting in L17 into three sentences. The relevant text now looks like this:

(P2 L16) "An example of this is the "cloud and precipitation feature database" of Liu et al. (2008). The database was derived from observations by the precipitation radar (PR), the Tropical Rainfall Measuring Mission (TRMM) Microwave Imager (TMI), the Visible and Infrared Scanner (VIRS), and the Lightning Imaging System (LIS) aboard the TRMM satellite. The authors performed several case studies with this dataset that contrasted continental and oceanic precipitating cloud systems, and found that …"

*P2 L20: 'larger extent' - please state whether horizontal or vertical extent is meant*

Horizontal extent. We have now clarified.

*P2 L26: Please specify references to the datasets used*

We added the references. We also provide references to the datasets later in section 2.

*P2 L33: 'examines' instead of 'examine'*

Done.

*P3 L3: 'Is there a closer' instead of 'Is there more close'*

Fixed.

*P3 L11: Please provide references for the MODIS instrument and specify the exact name of the dataset.*

In the sentence that immediately follows we provided such a reference "The MODIS cloud dataset (King et al., 2003) provides Level-3 …" We also added Platnick et al. (2003) and now call out the MODIS cloud dataset specifically.

(P3 L16) "The MODIS cloud dataset (MOD08_D3 and MYD08_D3; King et al., 2003; Platnick et al., 2003) provides Level-3 cloud products at daily time scales with 1°×1° horizontal resolution."

*P3 L22: Please rephrase and/or explain why a lower number of bins was chosen.*

The reason of reducing dimension is provided in subsection 2.3 when we explain the correlation method. Without coarsening we don't have correspondence with the ISCCP cloud types. The added sentence reads as follows:

(P3 L26) "In this study, the joint histogram bins are coarsened from 42 bins to 9 cloud types because of practical considerations (see subsection 2.3) as well as our desire to draw an analogy with the ISCCP cloud types (Chen et al., 2000; Rossow and Schiffer, 1999)."

*P3 L26: leave out 'best'*

We just follow the original expression in Huffman et al. (2007): "…, with the goal that the final product will have a calibration traceable back to the single "best" satellite estimate." Our intention was to convey the philosophy of TMPA.

*P4 L1: Specify the overpass time of MODIS Terra and Aqua in local time / equator crossing time.*

The equator crossing time, 10:30am for Terra and 1:30pm for Aqua (LST) has already been provided in the sentence.

*P4 L14: Please rephrase part with 'algorithmic variations'*

Changed to:

(P4 L20) "by these algorithm differences"

*P4 L19: Please rephrase the first part of the sentence*

The first part of the sentence has been re-written as follows:

(P4 L25) "Because the 3B42 dataset has higher spatial resolution than the MODIS Level-3 cloud dataset, we resample it to the…"

*P4 L21-22: Use consistent format of grid resolution, i.e. either 1° or 1x1°*

We now use 1°×1° throughout.

*P4 L24: 'of a histogram', instead of 'of histogram'*

Fixed.

*P4 L24: 'without missing values' instead of 'when no missing values exist'*

Thank you for suggestion, but it is actually rephrased as:

(P4 L31) "when there are no missing values"

*P4 L32: 'shows the distribution' instead of 'shows distribution'*

Fixed.

*P5 L3: Please rephrase the sentence, for example: '... the TMPA and MODIS observations also need to be matched in time.'*

Thank you, it does indeed sound better as suggested.

(P5 L8) "…, the TMPA and MODIS observations also need to be matched in time."

*P5 L15: Please rephrase 'in explaining'*

The sentence has been re-written as follows:

(P5 L21) "Other differences in occurrence frequencies between original and matched data are probably due to the diurnal cycle of precipitation."

*P5 L16: Please rephrase 'For example' at the beginning of the sentence*

"For example" has been removed.

*P5 L17: What is meant exactly by: 'relatively suppressed'?*

Changed to "relatively weak".

*P5 L28: Please rephrase sentence starting with "This is simply ... " as this would have been probably possible and could, in fact, provide additional insight, but was not pursued for practical reasons.*

We rephrased as follows:

(P6 L3) "Analysis and visualization of such a large number of coefficients are impractical, hence we pursue an analysis where both the cloud and precipitation histograms are coarsened.

*P6 L4-7: Please rephrase last part of sentence starting at "with no confusion resulting"*

Rephrased as follows:

(P6 L16) "For simplicity, the same symbols are henceforth also used to represent the frequency of occurrence within these groups, since their meaning is always clear by the context."

*P6 L13: Please explain what it is meant by the co-variability of anomalies.*

This is a convoluted way to simply say "correlations" (calculated from deviations from the mean), so now we just say "correlations".

*P6 L27: Replace 'of no-rain case' with 'of the no-rain case'*

Changed as suggested.

*P7 L3: Please indicate in which section the issue of less rain over land is analysed.*

Done.

(P7 L16) "The issue of less rain over land is also covered in the next composite plots (Figs. 5 and 6)."

*P7 L4: Please make it clearer to reader what you mean by the 'composite mean cloud and precipitation histogram'*

The composite means are cloud and precipitation histograms that were conditionally averaged. The condition in this particular case was that at least one occurrence of P4 or P5 precipitation existed in the 1°×1° grid cell, as described in the line that follows.

*P7 L19: 'in the P4 group' instead of 'due to the P4 group'*

Rephrased as suggested.

*P8 L13 and L16: is the increase really linear*

"linearly" was replaced by "monotonically"

*P8 L29: Please rephrase 'a factor affecting the Fig. 7 results''*

Rephrased to:

(P9 L9) "…, may be affecting the land results of Fig. 7".

*P10 L6: Please rephrase 'the peak negative value is a weaker value of'*

Rewritten as follows:

(P10 L27) "…, the peak negative value weakens to -0.23 and …"

*P10 L24: Please rephrase the second part of the sentence to make clear what you mean exactly.*

Rephrased as follows for clarity:

(P11 L10) "…, but similar peak correlations over land occur even for Cs and Cb."

*P12 L8: Please rephrase 'but suffice it to say here'*

We don't think that this needs to be rephrased.

*P13 L11: Please rephrase the beginning of the sentence (not start with 'But')*

"But" is replaced by "However,"

---

## Author Comment (AC2) · 21 Dec 2017

*General comments:*

*In "Contrasting the Co-variability of Daytime Cloud and Precipitation over Tropical Land and Ocean", Jin et al. analyze satellite datasets of cloud and precipitation retrievals with the aim of determining the relationship between cloud and rain. The topic is highly relevant for understanding the behavior of the atmosphere, both in physical reality and in parameterized cloud and precipitation in models. To my knowledge, the authors are the first to use this particular technique of regime-based cloud classification to analyze the relationship between cloud and precipitation, and I therefore recommend the analysis be published. The authors have chosen a fairly non-straightforward analysis method, and I hope the comments below will help them clarify a few points for the reader.*

> We thank the reviewer for the overall positive assessment of our work and the recognition that we have introduced an original analysis approach.

*Major comments:*

*The main potential weaknesses of the analysis are the following:*

*1. As the authors themselves point out, using cloud optical thickness and cloud-top pressure to define cloud regimes is an essentially ad-hoc classification based on arbitrary choices. In the conclusions, they then describe the regime classification as "widely accepted". It is true that these regimes are widely used, subject to the known caveats that the authors correctly state (e.g., that the regime names are not to be taken to correspond literally to actual cloud types); however, this acceptance is based on the regimes' usefulness having been demonstrated for each particular application, for example by showing that susceptibilities to aerosol are very different across regimes (e.g., Gryspeerdt et al, ACP 2014). In my opinion, this paper provides some interesting indications that the regime classification does indeed differentiate between cloud type of very different behavior regarding precipitation, but this is the case for only three (out of nine) regimes, so I think it will take some extra work (perhaps the unpublished paper referred to in the Conclusions) before the field will "widely accept" the use of these regimes for precipitation studies.*

> We take the opportunity here to clarify that our work uses the concept of "cloud type" rather than that of "cloud regime" (aka "weather state") as in Oreopoulos et al. (2014, 2016), the Gryspeerdt work quoted above, and previously in Rossow et al. (2005), among others. This is an important distinction. The presence of cloud types in a grid cell is simply described by the fraction of pixels within appropriate CTP and COT boundaries. On the other hand, when discussing cloud regimes, each cloudy grid cell can belong only to one cloud regime (derived from clustering analysis), specifically the one whose centroid histogram minimizes the Euclidean distance from the grid cell's particular joint histogram occurrence. In other words, a regime

represents a mixture of cloud types, with usually one of them being dominant. A combined cloud regime-precipitation analysis was conducted by Lee et al. (2013) and was completely different in character than the present study. Had we used cloud regimes we could have conceivably correlated each grid cell's total CF with the values in the 16 different precipitation bins (or 5 precipitation groups), and composited the results by regime. But we chose instead the cloud type approach since there is greater familiarity in associating cloud types with precipitation. While cloud types have initially been defined in terms of cloud appearance as viewed from surface human observers, the ISCCP definition of assigning cloud types from space-based passive observations using detected cloud extinction and vertical location is also considered quite standard, see https://isccp.giss.nasa.gov/cloudtypes.html (an early version appeared in Rossow and Schiffer, BAMS 1991, Fig. 4). We hope that this background clarifies the use of the expression "widely-accepted concepts about how to classify clouds into various types from passive observations" in p. 16, line 15 that the reviewer seems to allude to. What is widely accepted is not our analysis method, but the particular definition of cloud types. (Or perhaps the reviewer's comment was prompted by our statement that "Our study aims to go beyond widely known cloud-precipitation associations…" in p. 2, line 26?) In any case, we appreciate the opportunity to provide clarifications on the distinction between cloud types and cloud regimes, both derived from the same CTP-COT histograms.

*2. Many of the conclusions are based on regime-composite Pearson correlation coefficients between cloud area fraction and precipitation intensity percentiles. The Pearson correlation coefficient is fraught with pitfalls. The authors would greatly assist the reader in his or her assessment of the robustness of the conclusions by providing:*

*(a) a representative scatter plot of the correlated variables in the case of a strong positive correlation and a strong negative correlation and*

*(b) a geographic map of correlation strengths for the strongly positively and negatively correlated cloud/precip categories to see, e.g., whether the subsidence regions, ITCZ, warm pool, SPCZ, and maritime continent contribute as expected to the global-mean positive and negative correlations.*

[Figure]

P4+P5 vs. Cb CF 2D Histogram [ExTP, Terra+Aqua ]

[Figure]

P4+P5 vs. Cu CF 2D Histogram [ExTP, Terra+Aqua ]

We thank the reviewer for the thoughtful comment that prompted us to conduct additional analysis. The figures above (now the supplementary Figs. 2 and 3) show 2-dimensional histograms of P4+P5 rainfall fraction and cloud type fraction co-occurrence (2D histograms are more appropriate than scatterplots given the large number of points). The upper panels are for *Cb*, and the lower panels for *Cu*. The samples are conditional to P4+P5>0, same as for the correlation shown in Figs. 8b or 9b. Gray color indicates very small percentage less than 0.01%, and white color indicates 0%. The histogram bin size is 1/16 (=6.25%), and the bin labeled as "50%" indicates bin boundaries from 46.875% to 53.125%. One can see that (for *Cb* CF=0 bin) heavy precipitation can occur at instances even when there is no *Cb* cloud, and that the probability of strong

precipitation is much higher for small or zero *Cu* fractions (which manifests as an anti-correlation in our correlation "pyramid" plots.

[Figure]

The above figures (also the supplementary Fig. 4) show correlation coefficients for each grid cell at 1°×1° scale. The regions of abnormally high or low correlation values (e.g., *Sc*-dominant regions, the Sahara, the Himalaya, etc.) usually have small sample size for both these cloud types. Positive correlation between heavy rainfall and *Cb* cloud appears independent of location when there are enough samples. The anti-correlation between heavy rainfall and Cu cloud is more notable in oceanic subsidence regions, and weaker over land or convective regions (e.g., warm pool region, ITCZ, SPCZ).

The relevant text added is as follows:

(P10 L10) "In order to get a sense of the physical reality represented by Pearson's *r,* we examined two-dimensional histograms of cloud type CF and P-group for both strong positive and strong negative correlations (Supplementary Figs. 2 and 3). We note that more samples are available for zero or small amount of cloud type fraction for each case, and the distribution patterns look otherwise reasonable. We also examined the geographical dependence of these correlations and found them generally insensitive to location (Supplementary Fig. 4)."

*3. According to the authors, the TMPA precipitation dataset uses cloud-top temperature to fill in precipitation information where radar is not available. Since cloud-top height information is also used in the regime definitions, I would expect some amount of potentially spurious correlation. Discussion of whether this effect has been considered would be appropriate in the text.*

This is a valid point. Spurious correlations may indeed arise due to the use of the IR information to identify both cloud type and estimate surface precipitation rates. Since there is a physical relationship between cloud top temperature and precipitation, it is difficult to disentangle the physical effect form the spurious effect.

However, we expect the spurious effect to be sufficiently tempered for two reasons. First, TMPA and MODIS use different sources (and wavelengths) of IR information. TMPA uses the microwave precipitation rates to calibrate IR brightness temperatures and establish spatially varying relationships to IR-based precipitation rates. On the other hand, MODIS uses longer wavelengths of IR data ("$CO_2$ slicing" for high and mid-level clouds producing most of the precipitation) for estimating cloud top altitude than those used for precipitation, which are window brightness temperatures. Moreover, cloud optical thickness, the other dimension used to define cloud type does not come from the IR, but from shorter, near-visible solar wavelengths. The fact that TMPA precipitation is based primarily on the (radar-calibrated) microwave observations, which in turn calibrate the IR brightness temperatures, helps a lot in making the datasets substantially independent.

**Minor comments:**

• Section 2.2, *"If the number of bins in the histogram is chosen to be also 16, each bin value falls between 0 and 1 in multiples of 1/16, the sum of all histogram bins at 1° grid cell is equal to 1, and sub-grid precipitation rates are thus converted to areal fractions of specific ranges of precipitation rates": I don't quite follow the why 16 is a magic number in the link between the number of bins and area fraction; since we end up with 6, not 16, bins in Sec. 2.3, are those not area fractions anymore?*

    The reviewer is correct, and 16 is indeed not a magic number. We correct the sentence accordingly.

    (P5 L1) "Hence, each bin value falls between 0 and 1 in multiples of 1/16, and sub-grid precipitation rates are interpreted as *areal fractions* of specific ranges of precipitation rates."

• *Fig. 5: define what is meant by "climatology".*

    We changed the caption of Fig. 5 to avoid the ambiguous term "climatology".

    "Conditional composite mean of 2D joint histogram of pc and τ (left column), differences from overall (unconditional) mean (middle column)…"

• *p. 11, first paragraph: I find the claimed link between P4, P5, and MCS tenuous; for example, if P5 indicates MCS (where we expect clouds at all levels), why are both Cb and Cs anticorrelated with low- and mid-level clouds?*

    According to our definition of *Cb*, it has high cloud top altitude with large optical thickness, which means that the cloud vertical structure is quite deep and probably extends to low altitude. Hence, at the 1°×1° grid level, more fraction of *Cb* means less fraction of all other cloud types. This is less true for *Cs* cloud, so it is more likely that *Cs* and low cloud co-exist at the same location than for *Cb*. However, the low cloud under the high cloud can remain undetected by MODIS due to the inherent limitations of passive sensors. Moreover, we should re-emphasize that the separation between *Cs* and *Cb* is quite inexact given the way these cloud types were defined by ISCCP.

• *Anticorrelation in the Cu case: I am surprised that Cu is so anti-correlated with rain; I always thought (perhaps my thinking is guided by the regime name, which the authors caution against) that this would be the regime that clouds with high in-cloud water content but low area fraction (hence low grid-scale optical thickness).*

    One advantage of using MODIS 2D joint histogram of cloud is that the data preserves the sensor's pixel level information at the grid scale, so no averaging (or interpolating) pixel information to grid level takes place. As

the reviewer realizes while heeding our advice for caution, the *Cu* cloud in our study is different from the *Cu* cloud that comes to mind in other situations (like the cumulus congestus he/she seems to visualize). The *Cu* cloud from MODIS joint histogram (according to ISCCP definitions) has low optical thickness even at small spatial scales (MODIS pixel resolution is approximately 1km). We added a sentence about this point.

(P6 L7) "While these cloud types were given the same names as the standard cloud types seen by human observers from the ground and have some affinity with them, they are only loosely connected with the widely recognized traditional cloud types."

• *Anticorrelation in the Cu case (still): It would be interesting to get to the bottom of whether this is a real effect (CAPE/stability) or shadowing artifact, and I think the authors could easily do it by looking at CloudSat profiles (since they are already using MODIS data, not much additional co-location would be needed). If it is an artifact, does that mean all of Fig. 8 could be simplified to just the first row of every 3×3 plot? (By the way, I think the matrix of additive/subtractive Pn>0 subsets in Figs. 8 and 9 is brilliant plotting strategy.) Anyway, my first guess at the source of the anti-correlation was open vs. closed-cell stratocumulus, and it was interesting to learn that that was not the reason.*

The obscuration effect for lower clouds definitely exists as does the tendency of certain cloud type combinations to not co-occur (combined effects are expressed by the results of Fig. 10). Still, the negative relationship between heavy precipitation and *Cu* cloud would likely remain even without these effects, because, as pointed out above, the *Cu* cloud in this study is defined as optically thin cloud with low cloud top which is not expected to precipitate. Given CloudSat's limitations in the detection of boundary layer and puny clouds (e.g., TAU < 3.6), especially under conditions of signal attenuation in the presence of *Cb* hydrometeors, we're not sure whether embarking to such an investigation would pay dividends. The 3×3 plot can probably not be simplified to just the first row because for weaker precipitations the CF of the high cloud types decreases so the likelihood of lower cloud obscuration also decreases. In other words, the degree of obscuration is not independent of the precipitation rate.

• *The other surprise for me in Fig. 8 is that cor(CF, f_prec) never goes above 0.6. All CBs precipitate, so I would expect the Cb CF should correlate much more strongly with f_prec. What am I missing?*

A main reason behind the apparently low correlation value is the non-rigorous definition of *Cb* from the joint histogram. As we showed in Fig. 3, when *Cb* cloud fraction is larger than 6.25% (= 1/16), the probability of any kind of precipitation is 0.9, which seems to be consistent to the reviewer's intuition. However, our correlation comes from area fraction of specific precipitation and cloud type at 1° grid scale. In nature, it is possible that heavy precipitation comes from clouds other than *Cb*, and it is also possible that (at least some part of) *Cb* cloud (as defined here) does not produce heavy precipitation. Furthermore, it is worth noting that, when we tested the same methodology with (temporally and spatially) higher resolution dataset (e.g., GPM-IMERG and CMORPH), we obtained higher (above 0.7) values for the correlation coefficients. So TMPA's spatiotemporal resolution may be dampening the correlations in this case.

• *Sec. 4: it should be clarified that the first paragraph is an aspirational statement about the cloud-physics field as a whole, since this study is an incremental advance*

We have rephrased to provide the clarity that the reviewer is seeking about the reach and scope of our work.

(P16 L7) "Even with such non-ideal data at hand, the community still aspires to answer fundamental questions such as: To what degree can precipitation be predicted given information about clouds? Conversely, with precipitation information at hand, can we provide good guesses about the nature of the clouds responsible? Is precipitation variability associated with cloud variability? Do answers to the above questions differ substantially between ocean and land? This paper seeks to contribute ideas and results that will help us make progress in obtaining concrete answers in the near future, especially if observations also make considerable strides."

• p.16 l.16: if "once detection of low clouds in the presence of high clouds and of warm rain over land improves" refers to the use of active rather than passive satellite sensors, the authors may be interested in Field and Heymsfield or Mulmenstadt et al (both 2015, GRL)

Thank you for the suggestion. We now cite these papers accordingly.

• p.16 l.20: The authors chose not to use L3 instead of L2 data, presumably for reasons of data management complexity. I don't think anyone would fault them for this choice, so the defensive tone of this sentence is out of place. Either that, or I misunderstood something about it.

The intention of the sentences in this paragraph is to contrast this study from others using L2 data. The sentence has been rephrased:

(P17 L7) "Our self-imposed objective to make the study general, multi-year, and applicable to half of the Earth's surface, led us to Level-3 gridded data as the most appropriate choice. While some of the details seen in previous studies that used Level-2 data will unavoidably be lost, our datasets are good enough to extract major features of cloud-precipitation co-variability and allow us to claim that they are broadly representative of this co-variability in the tropics."

• p. 16 l.25: No objection to citing unpublished work, but why not also some published references that show the same thing, e.g., Suzuki et al (2015, J Atmos Sci), Jing et al (2017, JGR)

The publication Tan et al. has been accepted, and we have updated the citation. Thank you for your other suggestions, which have now been added too.